



# A field intercomparison of three passive air samplers for gaseous mercury in ambient air

Attilio Naccarato[1]*, Antonella Tassone[1], Maria Martino[1], Sacha Moretti[1], Antonella Macagnano[2], Emiliano Zampetti[2], Paolo Papa[2], Joshua Avossa[2], Nicola Pirrone[1], Michelle Nerentorp[3], John Munthe[3], Ingvar Wängberg[3], Geoff W. Stupple[4], Carl P. J. Mitchell[5], Adam R. Martin[5], Alexandra Steffen[4], Diana Babi[6], Eric M. Prestbo[6], Francesca Sprovieri[1], Frank Wania[5]*

[1]CNR-Institute of Atmospheric Pollution Research, Division of Rende, UNICAL-Polifunzionale, I-87036 Arcavacata di Rende, CS, Italy;
[2]CNR-Institute of Atmospheric Pollution Research, Research Area of Rome 1, Via Salaria km 29,300, 00016 Monterotondo, Italy;
[3]IVL Swedish Environmental Research Institute, Gothenburg, 41133, Sweden;
[4]Air Quality Processes Research Section, Environment and Climate Change Canada, Toronto, M3H 5T4, Canada;
[5]Department of Physical and Environmental Sciences, University of Toronto Scarborough, Toronto, M1C 1A4, Canada;
[6]Tekran Instruments Corporation, 330 Nantucket Boulevard, Toronto, Ontario, M1P 2P4, Canada;

*Correspondence to*:
Attilio Naccarato (attilio.naccarato@iia.cnr.it)
Frank Wania (frank.wania@utoronto.ca)



## Abstract

Passive air samplers (PASs), providing time-averaged concentration of gaseous mercury over the time scale of weeks to months, are promising to fill a gap in the monitoring of atmospheric mercury worldwide. Their usefulness will depend on their ease-of-use and robustness under field conditions, their availability and affordability, and most notably, their ability to provide results of acceptable precision and accuracy. Here we describe a comparative evaluation of three PASs with respect to their ability to record precisely and accurately atmospheric background concentrations at sites in both southern Italy and southern Ontario. The study includes the CNR-PAS with gold nanoparticles as a sorbent, developed by the Italian National Research Council, the IVL-PAS using an activated carbon-coated disk, developed by the Swedish Environmental Research Institute, and the *Mer*PAS® using a sulfur-impregnated activated carbon sorbent, developed at the University of Toronto and commercialized by Tekran. Detection limits are deduced from the variability in the amount of mercury quantified in more than 20 field blank samples for each PAS. Analytical and sampling precision is quantified through 22 triplicated deployments for each PAS ranging in length from two to twelve weeks. Accuracy and bias are assessed through comparison with gaseous elemental mercury concentrations recorded by Tekran 2537 automated mercury analyzers operating alongside the PASs at both locations. The performance of the PASs was significantly better in Italy, with all of them providing concentrations that are not statistically significantly different from the average of the active sampling results. In Canada, where weather conditions were much harsher and more variable during the February through April deployment period, differences were observed amongst PASs. At both sites, the *Mer*PAS® is currently the most sensitive, precise and accurate among the three PASs. A key reason for this is the larger size and the radial configuration of the *Mer*PAS®, which results in blank levels that are lower relative to the sequestered amounts of mercury than in the other two PASs, which rely on axial diffusion geometries. Because the blank-correction becomes relatively smaller with longer deployment, sampler performance tends to be better during deployments of 8 and 12 weeks.



## 1. INTRODUCTION

Mercury (Hg) is a highly toxic pollutant, which due to its significant adverse impact on ecosystem and human health has been added to the environmental political agenda at national, regional, and global levels. In recent years, the adoption of the Minamata Convention has aimed to protect human health and the environment from Hg releases and emissions (UNEP, 2013). Article 22 of the Convention requires Parties to formally assess, through the provision of "comparable monitoring data on the presence and movement of mercury and mercury compounds in the environment", how effective the structure and implementation of the Convention is at meeting its primary goal (Article 1). Article 19 of the Convention highlights the importance of environmental monitoring. While such efforts should build on existing monitoring networks (UNEP, 2013), this will also require research and development of monitoring technologies.

The accurate assessment of air pollutants has increasingly come into focus as the need to understand their transport and mechanisms of deposition to ecosystems grows (Dinoi et al., 2017; Moretti et al., 2020; Naccarato et al., 2018, 2020; Tassone et al., 2020). Special attention is given to the atmosphere, because it is a well-recognized pathway for Hg distribution throughout various environmental compartments (Driscoll et al., 2013). In this context, many regional atmospheric networks have been operating since the mid-1990s, including the US National Atmospheric Deposition Network- Mercury Deposition Network (NADP-MDN) (Vermette et al., 1995), the Environment and Climate Change Canada Atmospheric Mercury Measurement Network (ECCC-AMM), the Arctic Monitoring and Assessment Programme (AMAP) group of long long-term measurements (Arctic Council, 1991) and the European Monitoring and Evaluation Programme (EMEP) (Tørseth et al., 2012). In 2010, the Global Mercury Observation System (GMOS) was created in an attempt to establish a global atmospheric Hg measurement network, integrating EMEP and AMAP, with more than 40 monitoring sites distributed worldwide. Since their beginning, there has been a growing interest in improving global monitoring of Hg by increasing the spatial resolution of gaseous Hg data especially in remote locations and in developing countries (Pirrone et al., 2013) in order to meet Minamata Convention objectives.

Current methodologies, however, have a limited ability to monitor Hg on a truly global scale. Indeed, the use of active automated sampling system based on sorbent traps with gold amalgamation, which are desorbed at relatively fine time-resolution (3-5 minutes) for Hg quantification (Brown et al., 2010; Landis et al., 2002; Munthe et al., 2001; Steffen et al., 2012;

Wängberg et al., 2001), may be limited by cost and the need for reliable electricity, consumables and maintenance by well-trained operators (Huang et al., 2014; McLagan et al., 2016b; Pirrone et al., 2013). Given these constraints, passive sampling has been proposed as a viable alternative or supplemental system to fill the gaps of worldwide Hg monitoring. Compared to active sampling instruments, passive air samplers (PASs) for gaseous Hg are relatively inexpensive and thus can be deployed in large numbers allowing for the identification and characterization of Hg sources through finely resolved spatial mapping

(Huang et al., 2014; McLagan et al., 2016b; Pirrone et al., 2013). PASs are also suitable for deployment at remote sites because they require no power supply and are based on the unassisted molecular diffusion of gaseous Hg. Moreover, they are easy to use, compact, and portable. In summary, the adoption of PASs raises the very real possibility of a sustainable, long-term global network of atmospheric Hg measurements that includes regions not covered by existing efforts.

  Over the past few years, a number of mercury PASs have been developed, each with different materials and

geometries (Macagnano et al., 2018; McLagan et al., 2016a; Wängberg et al., 2016). While each sampler has its merit, the performance of different designs has yet to be compared systematically. This remains an impediment for understanding which PASs may be most appropriate for possible adoption in monitoring networks or whether a mix of designs can be reliably employed.

  In this paper, for the first time, we report the results of a field-based inter-comparison campaign and a controlled,

blind performance comparison among different Hg PASs. Three different PASs and their performance were evaluated at two monitoring sites, located in Italy and Canada, over a three-month period. The PASs involved in this study were developed by the Italian Institute of Atmospheric Pollution Research (CNR-IIA) (Macagnano et al., 2018), the Swedish Environmental Research Institute (IVL) (Wängberg et al., 2016) and the University of Toronto (McLagan et al., 2016a). Data were submitted for compilation to a blind third party in order to control for bias. The performances of the PASs were assessed for accuracy

through comparison with active sampling data, for precision and for sensitivity (e.g. the method detection limit, MDL), as well as in terms of the linearity of uptake over extended deployment periods.


## 2. METHODS

### 2.1 Passive Air Samplers (PASs)

Characteristics of the three PAS designs included in the comparison are summarized in Table 1. The CNR-IIA PAS
(CNR-PAS) consists of a fibrous quartz filter coated with sorbent material, which is attached to the bottom of a borosilicate
glass vessel equipped with a double cap system to minimize operator handling and avoid contamination due to the cap opening
(Macagnano et al., 2018). The IVL-PAS consists of a disk coated with an activated carbon sorbent that is inserted in a badge-
type device (Wängberg et al., 2016). The geometry of CNR-PAS and IVL-PAS makes them both axial diffusion samplers. The
*Mer*PAS®, developed at the University of Toronto and commercialized by Tekran Instrument Corp., consists of sulfur-
impregnated activated carbon sorbent (HGR-AC, Calgon Carbon Corp.), housed in a stainless-steel mesh cylinder that is
inserted into a commercial white Radiello® (Sigma Aldrich) diffusive body  (McLagan et al., 2016a). The *Mer*PAS® is a radial
sampler.

Table 1  Characteristics of the three passive air samplers for gaseous mercury that were compared in this study.

|  | **CNR-PAS**<br>Macagnano et al., 2018 | **IVL-PAS**<br>Wängberg et al., 2016 | *Mer*PAS®<br>McLagan et al., 2016a |
|---|---|---|---|
| Photograph | | | |
| Design principle | Axial diffusion badge | Axial diffusion badge | Radial diffusion |
| Sorbent material | TiO₂ nanoparticles, finely functionalized with smaller gold nanoparticles (reusable) | Activated carbon impregnated with 75 µl 0.1 % iodine solution (not reusable) | 0.6 grams of sulfur-impregnated activated carbon (HGR-AC) (not reusable) |
| Sorbent carrier | Fibrous quartz filter | Activated carbon and cellulose (Whatman) filters | stainless-steel mesh cylinder (reusable) |
| Diffusive barrier | Nylon membrane | Membrane FALP | White Radiello® diffusive body, porous high-density polyethylene (reusable) |



| Overall sampler dimensions | Height 3.1 cm<br>Diameter: 2.4 cm (w/o the cap) | Height: 1.2 cm<br>Diameter: 2.5 cm | Height 7.6 cm<br>Diameter: 7.2 cm |
|---|---|---|---|
| Effective diffusion pathlength (area) | 2.8 cm ($3.1 cm^2$) | 1.15 cm ($4.9 cm^2$) | 0.77 cm (without air boundary layer)<br>(~$7.5 cm^2$ on outer sorbent surface, ~$30 cm^2$ on diffuser surface) |
| Shelter | HDPP side shield with 8 seats used as PAS holder and for protection of the samplers from atmospheric agents and solar irradiation (reusable) | Metallic disc top, no side shield, 4 seats (reusable) | Each sampler is integrated within a compact PET protective shelter with a downward-facing mesh screened lid (reusable) |
| Deployment | Remove from double sealed aluminum bag, remove screw cap and store in bag while sampler snaps into broom holder style clip mounted to rain hood. Field Blank sampler never opened, installed with cap for duration of sample. | Open simple sealed bag, remove sampler from canister and snap into broom style clip mounted to rain hood. Travel blanks were taken to field site, but not opened. | Open simple sealed bag, remove tape from seal, replace solid screw cap with mesh screen cap, and install to mounting bracket with top threaded post and cap-nut. Field blank sampler never opened and installed with cap for duration of sample |
| Storage & Transport | Double cap seal, heat- and zip-sealed aluminum bags containing scrubber | plastic tube placed in a plastic bag | Integrated, compact PET protective shelter is used for storage and transport. Lid is tape sealed and sampler placed in plastic bag. |
| Hg Analysis method | Thermal desorption, gold amalgamation with CVAFS detection | Wet-digestion with chemical reduction, gas-liquid separation, gold amalgamation with CVAFS detection | Thermal desorption, gold amalgamation with CVAAS detection (USEPA Method 7473) |
| Analytical equipment | CNR-custom-build thermal desorption units interfaced with a Tekran 2537 mercury vapor analyzer | IVL-custom-build thermal desorption unit interfaced with a Tekran 2500 mercury vapor analyzer. | Commercially available automated total mercury analyzer (e.g. NIC MA-3000, Milestone DMA-80) |
| Sampling rate | $0.0147 \pm 0.0007$ $m^3$ $day^{-1}$ | $0.030 \pm 0.002$ $m^3$ $day^{-1}$ | $0.111 \pm 0.017$ $m^3$ $day^{-1}$ |

**2.2 Study Design**

The study design involved the side-by-side deployment of the three PAS types in the vicinity of existing active air

sampling sites in Rende (Italy) and Toronto (Canada) during late winter and early spring of 2019. At both sites, 11 overlapping

PAS deployment periods ranged in length from 2 to 12 weeks, whereby each deployment involved triplicate PASs and one

unexposed PAS as field blank, for a total of 88 PASs of each type. In some cases, additional storage blanks were taken. Each



participating research group supplied their PASs along with deployment instructions, performed the chemical analysis and

reported volumetric air concentrations and basic QA/QC results to an independent third party. Gaseous elemental mercury

(GEM) concentrations recorded by active air sampling instruments, averaged for the 22 deployment periods, were reported at

the same time. After data submission only the following changes were made to the data: A typographical error affecting the

assumed uncertainty of the sampling rate of the IVL-PAS was corrected. The blank correction for the CNR-PAS was performed

using the average of the field blanks at one location instead of using field blanks specific for a deployment, in order to be

consistent with the blank correction applied for the other two PASs. Results for the *Mer*PAS® with temperature-adjusted

sampling rates, that had also been submitted were disregarded, as they were the only set of results seeking to take into account

this effect. Temperature adjustment did not improve the accuracy of the the the *Mer*PAS® results. Although the field sites were in

Canada and Italy, respectively, none of the groups reporting the results for the PASs had prior knowledge of the results from

the active measurement at either location.

**2.3 Sampling Sites**

Two monitoring sites were selected for the performance evaluation of the three different PASs. The Italian sampling

site was a monitoring station close to the CNR Institute of Atmospheric Pollution Research (39°21'27.2"N 16°13'53.7"E) in

Rende. The Canadian sampling site was located on the grounds of the Downsview office of Environment and Climate Change

Canada, in a Northern suburb of Toronto (43°46'49.65"N 79°28'2.46"W). During the campaign, air temperature, relative

humidity, wind speed, and wind direction were measured at both sites. In Rende, meteorological data were recorded at a

meteorological station situated at few meters from the deployment area that was equipped with a thermo-hygrometer (LSI

LASTEM DMA875) for the monitoring of temperature and relative humidity, a pluviometer (LSI LASTEM DQA030) for the

acquisition of precipitation depth and an anemometer (LSI LASTEM DNA821) for the acquisition of wind speed and direction.

In Rende, the weather conditions were characterized by an average temperature of 12.0 ± 4.6 °C (range 0.9 – 30.6 °C) and

average relative humidity of 60.4 ± 18.1 % (range 13.6 – 97.9 %). Total rainfall over the study period was 1.71 mm and no

precipitation fell as snow. The wind blew mainly from SSE with an average speed of 1.2 ± 0.9 m s$^{-1}$ (range 0 and 6.9 m s$^{-1}$).

Meteorological data in Toronto were obtained from a co-located integrated weather station (Vaisala WXT520) operated by the

Ontario Ministry of Environment, Conservation and Parks. Temperature was variable through the spring season with a mean temperature was $0.6 \pm 6.6$ °C (range -15.8 – 21.5 °C), wind speed averaged 2.4 m s$^{-1}$ (range of hourly averages 0 to 11 m s$^{-1}$)

and average relative humidity (based on raw, uncorrected data) of $64.7 \pm 15.7$ % (range 18.5 – 92.1 %). The predominant (20.5 % of the time) wind vector was from the west, between 260 and 285°. Total precipitation during the PAS deployment period was 225 mm, of which 168 mm fell as rain and the remainder as snow.

### 2.4 Active Air Sampling

At both sites, gaseous mercury concentrations were obtained at 5-minute intervals using Tekran 2537x and 2537a

automated mercury analyzers (Tekran Instruments Corporation, Toronto, ON, Canada). In Toronto, two systems, namely a Tekran 2537x and 2537a (5037 and 0075 units, respectively) were operating in parallel in order to be able to quantify the duplicate precision of the active air sampling technique. These systems collect air onto gold traps, which are thermally desorbed for quantification of mercury by atomic fluorescence spectroscopy (Ebinghaus et al., 2011). The sampling was performed with airflow rates of 1.5 and 1.0 L min$^{-1}$ at Rende and Toronto, respectively. To ensure that Tekran systems were operating

consistently, flow verifications and calibrations were carried out before and during the intercomparison campaign by external injections of mercury and by using the instrument's internal mercury permeation source for automatic calibration at 23-hour and 72-hour intervals in Toronto and Rende respectively, throughout the monitoring period.

Calibration results and data acquisition were quality controlled according to established quality assurance and quality control procedures (QA/QC). The GMOS-Data Quality Management (G-DQM) (D'Amore et al., 2015) was used to check the

Tekran 2537x mercury concentration data collected at the Rende site and to monitor the performance of the instrument in terms of baseline shifts, sample volume cell bias and difference between gold traps, thus verifying that it adhered to standard procedures, in a way that minimizes losses and inaccuracies in data production. The Toronto QA/QC system used to check all data collected by Tekran analyzers was based on the Research Data Management Quality (RDMQ) standards defined in (Steffen et al., 2012). These standards invalidate data based on cell bias and sample volume, while also monitoring for baseline

shifts and deviation amongst other warning flags.

**2.5 PAS Deployment**

Samplers were sent by international courier from each participating laboratory to the two sampling locations shortly before the first deployment period. Following instructions provided by each participating research group, the samplers were deployed on a metal support rack at a height of about 4 m above ground to facilitate free air circulation (Fig. S1). At both sites,

all PASs were within 2 m of each other and from the inlet of the active air sampler. When not deployed, samplers were stored on-site at room temperature. Samplers were returned to the participating laboratories, again by international courier, shortly after the end of the last deployment.

While the three PASs were treated the same as much as possible, there were some unavoidable differences. The IVL-PASs made return air trips by international courier to both sampling sites and were deployed at both sites by personnel with

no experience with this sampler. The CNR-PAS did not need to undergo extended travel to the Rende site and the *Mer*PAS® was only transported by car between different locations within the city of Toronto (Tekran facilities, University of Toronto Scarborough Campus, ECCC sampling site in Downsview). At Rende, CNR-PASs were deployed by personnel with some familiarity working with this sampler; the same occurred at Toronto with the *Mer*PAS®. In both locations, several personnel were involved in the deployment and retrieval of PASs over the 12 weeks of the study, but it was always the same personnel

that handled all three PASs at any one of the seven deployment and retrieval dates.

After removal of the top cap, CNR-PASs and IVL-PASs were positioned in the seats of the shelter with the diffusive membrane or steel mesh net facing downwards. After exposure, CNR-PASs were removed from the seat, closed with the top cap, and placed into an aluminum bag containing a mercury scrubber cartridge. IVL-PASs were similarly removed, placed in a plastic container and then in a plastic bag. *Mer*PAS® samplers were secured to the metallic support using the embedded screw

and bolt dowel and the solid lid was replaced with the screened lid. After exposure, the screened lid was replaced by a solid lid, and the device sealed with tape and placed in a Ziploc bag.

PASs were deployed from February 5th to April 30th, 2019, following a sampling plan that included four deployments of 2 weeks, three deployments of 4 weeks, two deployments of 6 weeks, and one deployment each of 8 and 12 weeks (Table S1). All PAS deployments were in triplicate, with the addition of a field blank for each type of PAS to check the potential for

contamination during transport, storage and handling of the samplers. The CNR-PAS and *Mer*PAS® field blanks were deployed

in the field alongside each triplicate without opening their cap or lid. The IVL-PAS field blanks were not deployed at the actual field locations, but were only briefly transported to the deployment site during a sample change-over. During the remainder of the 12 weeks of the study, they stayed in storage in indoor locations in Rende and Toronto. Additionally, five storage blanks of the CNR-PAS in each of Rende and Toronto were used to check for mercury contamination during the PAS storage and

transport. There were only five such storage blanks of the *Mer*PAS® in Rende, i.e., none in Toronto.

## 2.6 Analysis of PAS Sorbents for Hg

Mercury in CNR-PASs was quantified using a CNR-IIA-designed thermal desorption system, comprising a glass cylinder housed in a heater furnace, connected to a mercury vapor analyzer (Tekran 2537a) for Hg detection by CVAFS. The sorbent membrane is placed into the cylinder, which is heated to 550 °C to desorb the trapped mercury (Macagnano et al.,

2018). After analysis, the collection surface of the CNR-PASs can be regenerated to be reused (Macagnano et al., 2018). The accuracy of the lab-made analytical system was periodically verified using CNR-PAS loaded with a known Hg concentration, while the Tekran system was calibrated by automatic and manual procedures.

For mercury determination of IVL-PASs the carbon filters were carefully removed from each sample and individually boiled in an acid solution ($HNO_3$/$H_2SO_4$) for 5 – 6 hours. BrCl was added to the cold solution as an oxidant and subsequent

reduction was performed by adding $SnCl_2$ prior to analysis. Excessive BrCl was reduced using hydroxylamine hydrochloride prior to addition of $SnCl_2$. Liquid-gas separation was performed using a purge system with Hg pre-concentration on a gold trap. The sample gold-trap was analyzed in an IVL-custom made desorption system connected to a CVAFS detector (Tekran 2500 unit) (Wängberg et al., 2016).

Determination of mercury concentration in activated carbon sorbent used in the *Mer*PAS® was carried out at the

Tekran laboratory in Toronto using a Nippon MA-3000 system for automated combustion, amalgamation, and detection by atomic absorption spectroscopy. Throughout the analysis, standard reference materials and liquid Hg standards (2 to 8 ng) added to activated carbon were analyzed. Standard reference materials were bituminous coal (NIST 2684b, NIST 2685) and an activated carbon sample generated in-house at the University of Toronto.

For QA/QC of mercury analytical data during sorbent analyses, both analytical and field blanks were used. Analytical

blanks were analyzed before deployment and sampling to ensure sorbent materials (HGR-AC, AuNPs-TiO$_2$NPs layer, and

activated carbon layer for *Mer*PAS®, CNR-PAS, and IVL-PAS, respectively) were free from Hg contamination. The field

blanks were used to ascertain whether there was contamination during sampler assembly, shipping, transport, deployment,

retrieval and storage. Storage blanks were used to assess any contamination due to the transport and storage only, i.e. not

during the handling of the PASs during deployment and retrieval operations.

The samplers were deployed in triplicate during the campaign to assess the precision of each PAS. Method detection

limits (MDLs) and practical quantification limits (PQLs) in ng were calculated as three and ten times the standard deviation

of the amount of mercury in field blanks, respectively. The limits of detection (LOD) and quantification (LOQ) in ng m$^{-3}$ were

obtained by dividing MDL and PQL by the product of sampling rate (*SR*) and deployment time (days).

**2.7 Determination of Volumetric Hg Concentration**

The average Hg concentration in the atmosphere measured by each sampler (C; ng m$^{-3}$) was obtained from the

analyzed mass of Hg in the sorbent material according to Eq. (1):

$$C = \frac{m}{t \times SR} \qquad (1)$$

where *m* is the mass of sorbed Hg (ng) corrected for the blank contamination, *t* is the deployment time of the PAS (days) and

*SR* is the sampling rate of the PAS (m$^3$ day$^{-1}$). Constant and previously experimentally derived *SR* values were used for each

PAS. For the CNR-PASs, the *SR* was 0.0147 m$^3$ day$^{-1}$ with an uncertainty of 0.0007 m$^3$ day$^{-1}$. This value is slightly different

from a previously reported one (Macagnano et al., 2018) because it is the result of further improvements of CNR-PAS

geometry. For the IVL-PASs, the *SR* was 0.028 m$^3$ day$^{-1}$ in Rende and 0.029 m$^3$ day$^{-1}$ in Toronto (calculated using the

diffusivity for Hg according to Massman (Massman, 1999)). The *SR* of the *Mer*PAS® (0.111 ± 0.017 m$^3$ day$^{-1}$) is higher than

that of the other two PASs, which is a function of the *Mer*PAS®'s radial design. This *SR,* which was derived from a number of

calibration experiments conducted by Tekran, deviates slightly from previously published values (McLagan et al., 2016a,





2018), because of small modifications between the *Mer*PAS® and the original sampler. For each PAS type, the uncertainty of

the *SR* is directly propagated to the volumetric air concentration.

## 2.8 Statistical Analysis

All statistical analyses were performed using R v. 3.3.3 software (R Foundations for Statistical Computing, Vienna,

Austria). We evaluated the relative accuracy of different PASs, by first calculating the percentage concentration differences

between actively sampled concentrations [Hg]$_{Tekran}$, and those derived from each of the paired PASs [Hg]$_{PAS}$. These percent

concentration differences were calculated as:

$$\%difference = \left(\frac{[Hg]_{Tekran} - [Hg]_{PAS}}{[Hg]_{Tekran}}\right) * 100 \tag{2}$$

Based on these calculations, we then used absolute concentration differences for subsequent analysis. First, we used a variance

partitioning analysis to quantify the proportion of the overall variability in absolute percentage concentration difference values

(calculated in Eq. 2), that is explained by 1) deployment site, 2) deployment time, 3) Tekran identity (in the case of Toronto),

and 4) PAS type. This variance partitioning analysis was based on *n*=99 total observations of absolute percent concentration

differences. To perform this analysis, we first fit a linear mixed effects model to our data using the '*lme*' function in the '*nlme*'

R package (Pinheiro J, Bates D, DebRoy S, 2017); in this model, absolute percent concentration differences are predicted as a

function of a single fixed effect (i.e., the model intercept, which represents the overall mean percent concentration difference),

and four random effects (i.e., four nested factors including 1) PAS type, within 2) Tekran IDs (alternatively, the deployment

location), within 3) deployment periods, within 4) deployment site (i.e., one of Rende or Toronto). Based on this model, we

then used the '*varcomp*' function in the '*ape*' R package (Paradis et al., 2004) to quantify the proportion of variation in

concentration differences that owes to each of the four nested factors.

Based on these results, we then sought to calculate and compare mean absolute concentration percentage differences

across both PAS types and sites, while accounting for 1) the non-independence of samples, 2) unbalanced sample sizes across

sites and PASs, and 3) potentially confounding effects of a) sampling deployment times and b) sites. Therefore, we

parameterized a second linear mixed effects model where absolute concentration differences were predicted as a function of

PAS type, site, and a PAS-by-site interaction term as fixed factors; this mixed model statistically accounted for non-

independence of samples, by including deployment period and Tekran identity as nested random effects. Based on this model,

we then used the '*lsmeans*' and '*difflsmeans*' functions in the '*lmerTest*' R package (Kuznetsova et al., 2017), to calculate and

statistically compared least square mean concentration difference values (and associated standard errors) across each PAS

type, site, and each PAS-by-site combination. This analysis therefore allowed to assess whether least square mean

concentration differences values in any of these groups, differed significantly from one another, or differed significantly from

zero.

## 3. RESULTS AND DISCUSSION

### 3.1 Mercury concentrations obtained by active sampling

The concentration of gaseous mercury in ambient air was determined by averaging the values recorded by the Tekran

Hg analyzers every 5-minutes during a specific PAS deployment period. The complete series of valid Hg concentration data

is displayed in Fig. S2 in which the interruptions to the sampling are due to instrument calibration or maintenance. As

mentioned in section 2.4, the values obtained at Rende were validated against the GMOS-Data Quality Management (G-DQM),

resulting in 98.9 % of valid data. The average measured Hg concentration at Rende over the 12 weeks was $1.72 \pm 0.25$ ng m$^{-3}$, with a range from 0.88 to 8.80 ng m$^{-3}$. The mean Hg concentration during the 11 PAS deployment periods was quite constant,

varying between 1.66 ng m$^{-3}$ and 1.79 ng m$^{-3}$.

At Toronto, the use of the RDMQ standards for data quality assessment and measurement gaps during daily

calibration periods, hourly standard additions and instrument maintenance resulted in 82.5 % of valid data coverage throughout

the entire deployment period for the primary 2537x analyzer. The secondary co-located 2537a analyzer experienced an 8 %

shift in the mass-flow meter calibration during the study. Since it was not possible to determine when the shift occurred, data

from this analyzer were not used for comparison with the PAS (but were included in the statistical analysis described in 2.8).

The active Hg concentration ranged between 1.17 and 34.6 ng m$^{-3}$ and averaged at 1.57 ng m$^{-3}$ with a standard deviation of

0.45 ng m$^{-3}$. Five short periods of elevated concentrations (over 4 ng m$^{-3}$) were observed over the study period, the maximum

reaching 34.6 ng m$^{-3}$. Although unusual, the elevated values were observed on both the primary and secondary analyzers and lasted between 10 and 35 minutes and are accepted as valid. The Toronto site is located in a northern suburb of Canada's largest urban center and it is believed that these elevated episodes are a result of nearby industrial mercury emissions. Similar

to Rende, the mean Hg concentration for each deployment varied only slightly, between 1.51 ng m$^{-3}$ and 1.63 ng m$^{-3}$.

## 3.2 Comparison of Passive Air Sampler Performance

### 3.2.1 Blanks and Detection Limits

The amount of Hg in the field blanks of the different passive air samplers are summarized in Table S2. The averages of those values are displayed in the top row of panels in Fig. 1. The amounts in field blanks are similar between the different

passive samplers, ranging from generally less than 0.2 ng in the CNR-PAS to slightly above 0.4 ng in the IVL-PAS. The blank levels of the CNR-PAS are the lowest recorded during the campaign, especially for exposure in Rende. The *Mer*PAS® showed no difference in the blank levels between Rende and Toronto, whereas the CNR-PAS and IVL-PAS showed a slight difference between the two sites. In the case of the *Mer*PAS® and CNR-PAS where field blanks were deployed together with samplers in the field for variable lengths of time, there was no indication that the field blank contamination increased with increasing time

in the field (Table S2). This is consistent with blank contamination arising during handling and transport and not during the placement at the deployment location. However, in the case of the CNR-PAS, the storage blanks (0.066 ng ± 0.010 ng, n = 5, Toronto; 0.042 ng ± 0.009 ng, n = 5, Rende) have, in general, considerably lower levels than the field blanks (0.20 ng ± 0.07 ng, n = 5, Toronto; 0.15 ng ± 0.02 ng, n = 5, Rende), which implies that the deployment and retrieval of those samplers does introduce some contamination. The amount quantified in *Mer*PAS® storage blanks (0.187 ng ± 0.009 ng, n=5, Rende) is only

marginally lower than the amount in field blanks (0.23 ng ± 0.06 ng, n=10, Rende).

The relative standard deviation (RSD) of levels in field blanks was also similar between the three samplers, being slightly lower in the IVL-PAS (~18 %) than in the *Mer*PAS® and CNR-PAS (~23 % on average). This may be a result of the IVL-PAS field blanks all being treated the same, whereas the *Mer*PAS® and CNR-PAS field blanks had slightly different handling processes, as they were deployed alongside the exposed samplers. The RSD of the CNR-PAS deployed in Rende was

notably lower (12 %) than in Toronto (34 %).

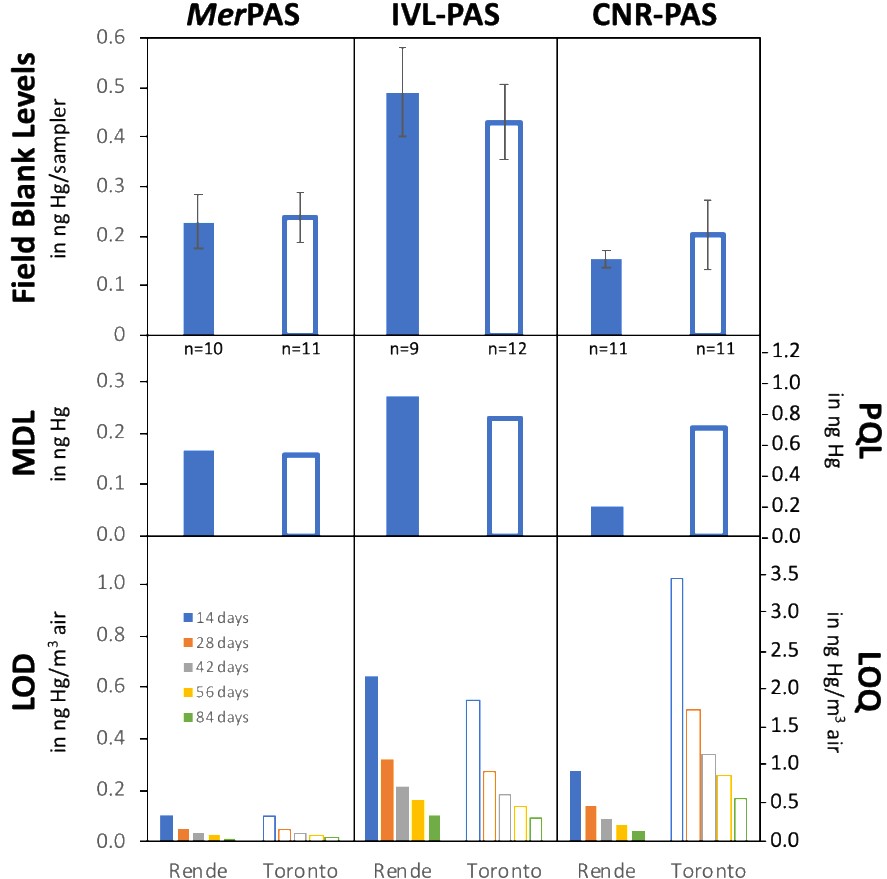

**Figure 1** Mean and standard deviation of field blank levels, method detection limit (MDL), practical quantification limit (PQL), limit of detection (LOD) and quantification (LOQ) for the three passive air samplers deployed in Rende, Italy and Toronto, Canada.

The amount of mercury detected in field blanks was used for the calculation of the method detection limit (MDL), the practical quantification limit (PQL), the limit of detection (LOD) and limit of quantification (LOQ). Field blank levels, MDLs, PQLs, LODs and LOQs for the three samplers separated for the two locations are displayed in Fig. 1. The numerical results can be found in Table S3 in the supporting information. The MDL and PQL are derived from the variability in the field blank levels. Therefore, they are similar between the three samplers (middle row of panels in Fig. 1). Even though the RSD of

the field blank levels is smaller for the IVL-PAS, the larger absolute SD means that it has slightly higher MDL and PQL (~0.25

ng and ~0.83 ng, respectively) than the other two samplers (0.13 and 0.45 ng for the CNR-PAS on average, 0.16 and 0.54 ng for the *Mer*PAS®).

In terms of volumetric air concentrations, the LODs and LOQs decrease with the sampled air volume, which, in turn, increases with a sampler's *SR* and deployment period. The bottom row of panels in Fig. 1 therefore displays the LODs and

LOQs for each of the five deployment times used in this study. Larger differences between the *Mer*PAS® and the other two PASs become apparent, because the former has a *SR* that is ~4 to ~8 times higher than that of the latter and accordingly samples Hg from a much larger air volume during similar deployment times. This implies that even though the absolute amounts of Hg in field blanks are similar between the samplers, the amount in field blanks relative to the amounts in exposed samplers is quite different (Table S4). Field blank contamination in the *Mer*PAS® does not exceed 8% of the quantified amount in an exposed

sampler (range 1 to 8 %), whereas that percentage in the IVL and CNR-PAS is similar and ranged between 7- 47 %, with higher values during short periods of deployment. This means that during a two-week deployment the *Mer*PAS® has a LOD of 0.11 ng m$^{-3}$ and a LOQ of 0.34 ng m$^{-3}$, which are ~6 times lower than the LODs and LOQs of the IVL-PAS and the CNR-PAS. It is important to stress that blank contamination and therefore also MDL/PQL and LOD/LOQ are study-specific and therefore need to be determined during every study anew. The two deployments on the CNR-PAS in Toronto and Rende

illustrate this very effectively. The lower and more consistent blank levels of the CNR-PASs deployed in Rende compared to those deployed in Toronto, translate into four times lower LODs (0.28 ng m$^{-3}$ vs. 1.02 ng m$^{-3}$ for a two-week deployment) and LOQs (0.9 ng m$^{-3}$ vs. 3.4 ng m$^{-3}$ for a two-week deployment).

### 3.2.2 Precision

The very large number of triplicate deployments in this study allows for a thorough characterization of the precision

of the different PASs. Specifically, we assess the replicate precision of three PASs deployed simultaneously, both before and after blank correction. Table S5 reports the amount of Hg quantified in the PAS during the 22 different deployments. The precision of the quantified amount in a PAS reported in this table is a combined measure of the consistency and reproducibility of PAS manufacturing, deployment and handling as well as the laboratory analytical process. Table S6 reports the blank-



adjusted amount of Hg in the PASs. The precision of the blank-corrected amounts reported in this table additionally accounts

for the consistency and reproducibility of the blank contamination.

The relative standard deviation in percent (RSD%) of the mean of the amount of Hg quantified in three samplers is

used as a measure of precision. Blank correction was performed using the average value of all field blanks deployed at one

location, because the field blanks did not show a dependence on deployment length for any sampler but did display differences

between Rende and Toronto deployments for some samplers. The precision of the blank-corrected amount was calculated by

propagating the standard deviations of the amount in exposed samplers and of the amounts in field blanks. Fig. 2 displays the

replicate precision for the three samplers, averaged for different deployment lengths, across the two location and across all

replicated deployments. Numerical results are presented in Table S7.

When judged based on the amount of Hg quantified in triplicated samplers, $Mer$PAS®, IVL-PAS and CNR-PAS had

an average precision across all 22 replicated deployments of 3 %, 9 % and 7 %, respectively. In the case of the $Mer$PAS® this

is consistent with previously reported replicated precision, e.g. 3.6 % in a global study involving deployments in numerous

location (McLagan et al., 2018). Replicate precision was generally similar in Rende and Toronto deployments, only the IVL-

PAS had on average slightly lower precision in Rende (~11 %) than in Toronto (~ 7 %). The replicate precision of the $Mer$PAS®

improved slightly with increasing deployment length (from ~5 % for the 2-week samples to ~2 % for the 12-week samples).

In general, one might expect larger amounts of Hg to be quantified more reliably than smaller amounts, which would explain

such a trend. The $Mer$PAS®, for example, collected ca. 3 ng of Hg in a two-week period, but ca. 17 ng in a 12-week deployment.

The IVL-PAS shows such a trend of improving precision with longer deployment between the 4-week (~13 %) and the 12-

week samples (~5 %). However, the relatively good precision of the 2-week samples (~7 %) does not fit this pattern. The

precision of the CNR-PAS was not related to deployment length, with the poorest precision for the 8-week deployments (~13

%) and the best precision for the 4-week deployments (~5 %).

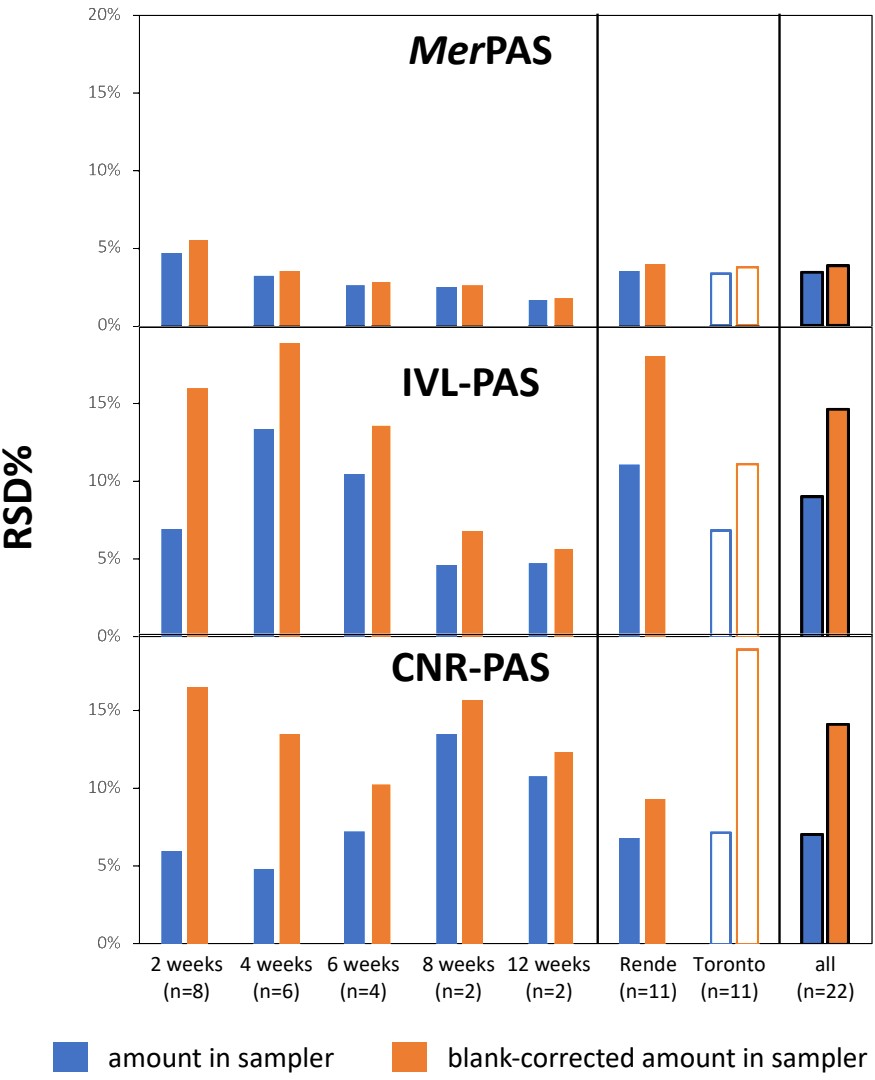

**Figure 2** Precision expressed as the relative standard deviation in percent of the amounts of mercury quantified in triplicate PASs, both before (blue) and after blank correction (orange), averaged over different deployment lengths, across different locations and over all replicated deployments. Note that in some cases, a sampler was lost and therefore some deployments were only duplicated.

When judged based on the blank-corrected amounts in replicate samplers, precision was 4 %, 15 % and 14 % for *Mer*PAS®, IVL-PAS and CNR-PAS, respectively. This precision is inevitably worse than for the non-blank corrected amount, because the variability of the field blank levels adds uncertainty. The extent of this increase in uncertainty upon blank correction depends very strongly on how large the blank contamination is relative to the amount in exposed samplers. This

explains why the increase is much smaller for the *Mer*PAS® than for the other two samplers and it also explains why the

increase is larger for shorter deployment periods. As was already mentioned above, the blank correction for IVL-PAS and

CNR-PAS deployed for 2 weeks is quite large, ranging from 28 to 47 % of the amount quantified in exposed samplers.

Therefore, the uncertainty of the deducted amount adds notably to the uncertainty of the blank-corrected value. However, the

CNR-PAS also illustrates how sampler precision can be greatly improved by consistent field blank levels. The blank correction

of the CNR-PASs deployed in Rende adds far less uncertainty (from 7 % to 9 %) than the blank-correction of the CNR-PASs

deployed in Toronto (from 7 % to 19 %), which is a result of the much smaller variability in the field blank levels measured

in Rende (See Fig. 1).

### 3.2.3 Accuracy

The average air concentration during each of the 22 deployment periods was derived by dividing the blank corrected

amounts in a PAS by the product of the deployment period and the *SR*. These concentration values are reported in Table S8.

The *SR* and its estimated uncertainty for each PAS was provided by each participating laboratory. Specifically, the uncertainty

of the *Mer*PAS® *SR* was assumed to be 15 %, whereas that of the CNR-PAS was 4.7 %. The *SR* uncertainty of the IVL-PAS

was assumed to be 6 % and 13 % during the deployments in Rende and Toronto, respectively. The uncertainty of the

concentration values in Table S8 was obtained by propagating the estimated uncertainty of the *SR* and the standard uncertainty

of the blank-corrected amounts (in Table S6). The average value of the relative uncertainty of the volumetric concentrations

is very similar between the three PASs: 9 % for *Mer*PAS® and CNR-PAS and 8 % for the IVL. However, these values cannot

be directly compared with each other, as the self-reported uncertainty of the SRs was not established the same way by the three

study participants.

The accuracy of the PAS-derived time-averaged air concentrations in Table S8 was judged by comparing them to the

average value derived by the active Tekran instruments, operating alongside the PAS. Tekran values were considered as a

benchmark for pragmatic reasons, knowing full well that this measurement itself may provide biased results (Aspmo et al.,

2005; Slemr et al., 2015; Temme et al., 2007), even though before, during and after flow and detector accuracy audits of the

active instruments were performed at both locations. The possible size of such bias was estimated from the data collected by





the two Tekran systems operating side-by-side in Toronto, although the comparison was somewhat hampered by an

inconsistency between measured flow at the beginning and end of the sampling period for the "0075" unit (see section 3.1). If

we disregard that uncertainty, the "0075" instrument yielded values averaged over the deployment periods that were

consistently lower than those measured by the "5037" instrument that was chosen as the reference. This bias was on average

3.2 % for the 11 sampling periods and ranged from a low of 1.0 % for the third 2-week period to a high of 6.5 % for the last

4-week period.

390       Table 2 summarizes the average bias and the average absolute difference between the average concentrations

measured by the Tekran 5037 instrument and the various PASs. This compilation reveals a number of features: The accuracy

of all three PASs is much better during the deployments in Rende than the deployments in Toronto. On average, the IVL-PAS

and CNR-PAS results for Rende show no bias, whereas the *Mer*PAS® results are slightly biased high (~3 %). Also, the absolute

discrepancies are quite small in Rende, averaging ~3 % for the *Mer*PAS® and ~7 % for the other two samplers. In Toronto, the

*Mer*PAS® air concentrations are biased high, on average 10 %. The IVL-PAS also shows a positive bias (~17 %), whereas the

CNR-PAS levels are on average 9 % lower than the Tekran results. The average, absolute discrepancies range from 10 % of

the *Mer*PAS® to 18 % for the IVL-PAS and 25 % for the CNR-PAS.

**Table 2** Average bias and average absolute discrepancy between the time-averaged volumetric air concentrations of Hg
      derived by passive air sampler and Tekran.

|  |  | *Mer*PAS® | IVL-PAS | CNR-PAS |
|---|---|---|---|---|
| Rende | Bias (%) | 2.8 | -0.5 | -1.4 |
| (n=11) | Absolute discrepancy (%) | 2.9 | 7.1 | 6.1 |
| Toronto | Bias (%) | 10.2 | 17.0 | -8.8 |
| (n=11) | Absolute discrepancy (%) | 10.2 | 17.8 | 24.9 |


      Figure 3 displays the discrepancies of the PAS results from the average concentrations measured by the Tekran

analyzers for each of the 22 sampling periods. This illustration reinforces the remarkable differences in the sampler accuracy





in Rende and Toronto. It additionally shows that there is no apparent relationship between the accuracy of the PASs and the

length of the deployment period in Rende. For the *Mer*PAS®, there is also no relationship between discrepancies and

deployment length in Toronto. The high bias tends to be consistent indicating that the *SR* was likely higher than the applied

value of 0.111 m$^3$ day$^{-1}$. The discrepancy of the IVL-PAS and CNR-PAS from the Tekran results tend to be smaller during the

longer deployments in Toronto (6 weeks and up). In fact, for the IVL-PAS the discrepancies tend to get smaller with increasing

deployment times. This makes sense considering that the uncertainty introduced by the blank-correction becomes much smaller

with longer deployments. The three 4-week deployments of the CNR-PAS in Toronto are consistently biased very low (by

about 50 %), whereas the two sampling periods with very high bias are both 2-week deployments, so, it is difficult to decipher

a consistent pattern in the discrepancies.

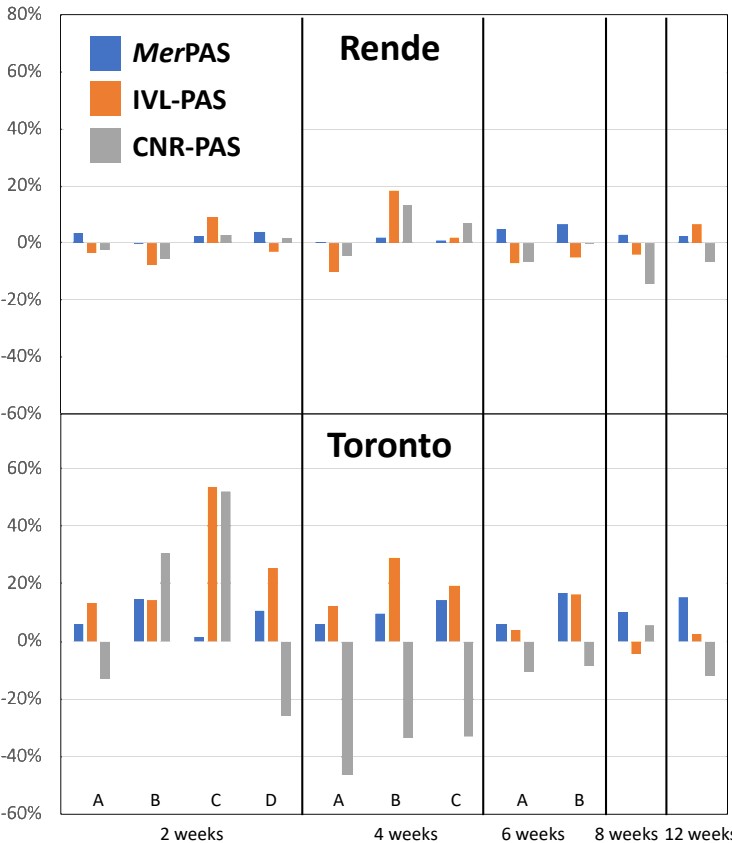

**Figure 3** Discrepancies of the time averaged air concentrations of Hg during 22 deployment periods as derived by the three
PAS from the average concentration obtained by an active Tekran system deployed at the same time. Deployments





in Rende/Toronto are displayed in the upper/lower panel, respectively. Positive/negative discrepancies indicate a
        PAS-derived concentration higher/lower than the Tekran value, respectively.

        Our variance partitioning analysis, coupled with mixed effects models, confirmed that all PAS-derived concentrations

        were significantly closer to the Tekran values for the deployments in Rende than they were for the deployments in Toronto

        (Fig. 4). The asterisks in Fig. 4 designate the significance level by which the mean absolute concentration difference of a

"dataset" differs significantly from 0, i.e. whether PAS-derived concentrations (based on Eq. 2) differ significantly from

        Tekran-concentrations. We can see that the mean concentration differences in Rende, for all three PASs individually, and when

        the data is "pooled" among all PASs, are not significantly different from 0. On the other hand, all of the mean concentration

        differences in Toronto site are significantly different from 0, again, for all PASs individually, and when the data is "pooled"

        among all PASs. When data from both sites and all PASs are "pooled" together, the mean concentration values differ

significantly from 0, which is mainly driven by the poorer agreement of values in Toronto. Note that we use here the terms

        "pooled" and "datasets", even though the results in Fig. 4 are based on the single mixed effects model, and are not the results

        of multiple $t$-tests.

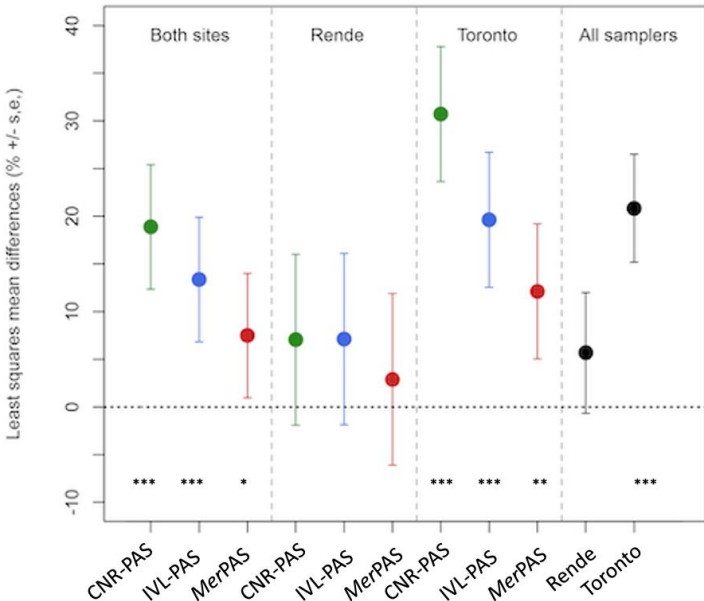

**Figure 4**  Least square means and standard errors of the differences in concentrations measured by the PASs and by the Tekran
430          units. Results are shown either for each PAS individually (colored markers) or for the three PAS together (black

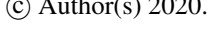



markers). They are also shown either separately for the two sampling sites or both of them together. The asterisks below each bar indicate whether or not the least square mean concentration differences from that "dataset" differed significantly from 0 (where *** denotes $p \leq 0.001$; ** denotes $p \leq 0.01$; and * denotes $p \leq 0.05$).

Of course, the same concentrations measured by different techniques should not be significantly different. It implies that the uncertainty of the concentrations derived from all three PASs deployed in Toronto must have been underestimated, i.e., the assumed uncertainty of the *SR* applied in the calculation of the concentrations must have been too small. We may surmise that if meteorological conditions during a deployment deviate considerably from those prevailing during the calibration of a PAS (as they did for all three PASs during a Toronto winter), the *SR* incurs considerably higher uncertainty,

than if calibration and application take place under similar environmental conditions.

The variance decomposition analysis attributed roughly half of the variance in percentage concentration differences to the PAS type (48.3 %) and most of the other half of total variance by differences observed between Toronto and Rende (site = 46.8 %) (Table S9). According to the mixed effects model, there were significant differences for both site (Rende vs. Toronto; p < 0.001) and PAS type (p = 0.006) (Table S10). In the post-hoc least squares comparison (Table S11), differences amongst

PAS types were not significant from one another at Rende (p = 0.458 to 0.992). At Toronto, the percentage difference between Tekran and PAS concentrations was not significant between the IVL and *Mer*PAS® samplers (p = 0.312), but both the IVL (p = 0.013) and *Mer*PAS® (p < 0.001) had concentrations significantly closer to Tekran values than did the CNR sampler.

### 3.2.4 Linearity of Uptake

A PAS's performance depends on having an uptake capacity that is sufficiently high for mercury to remain in a linear

uptake phase throughout the entire deployment period. We can test this by assessing the linearity of uptake. While this is sometimes done by plotting the blank-corrected amount quantified in the samplers $m_{PAS}$ against the sampler deployment time $\Delta t$, this disregards the variability in the GEM concentrations between different deployments. By plotting the amount in a sampler against the product of $\Delta t$ and the average air concentration during the deployment of that sampler $C_{air}$, we can eliminate the influence of the GEM concentration variability (Restrepo et al., 2015). We used the data from the Tekran instruments as

the input for $C_{air}$ in this analysis. Incidentally, using the data from this intercomparison study this way amounts to a sampler





calibration, as the slope of the linear relationship between $m_{PAS}$ in ng and $\Delta t \cdot C_{air}$ in units of days·ng m$^{-3}$ corresponds to the *SR*

of the PAS in m$^3$ day$^{-1}$.

Figure 5 shows these uptake plots for all three samplers at the two sampling locations. Also shown are the linear

regression lines fitted to the displayed data. Table 3 reports the slopes with standard error of the regression line, which has

been forced through the origin, and the coefficient of correlation $r^2$. The slopes are also the *SR* applicable to the PASs at the

two locations during the time period of the study.

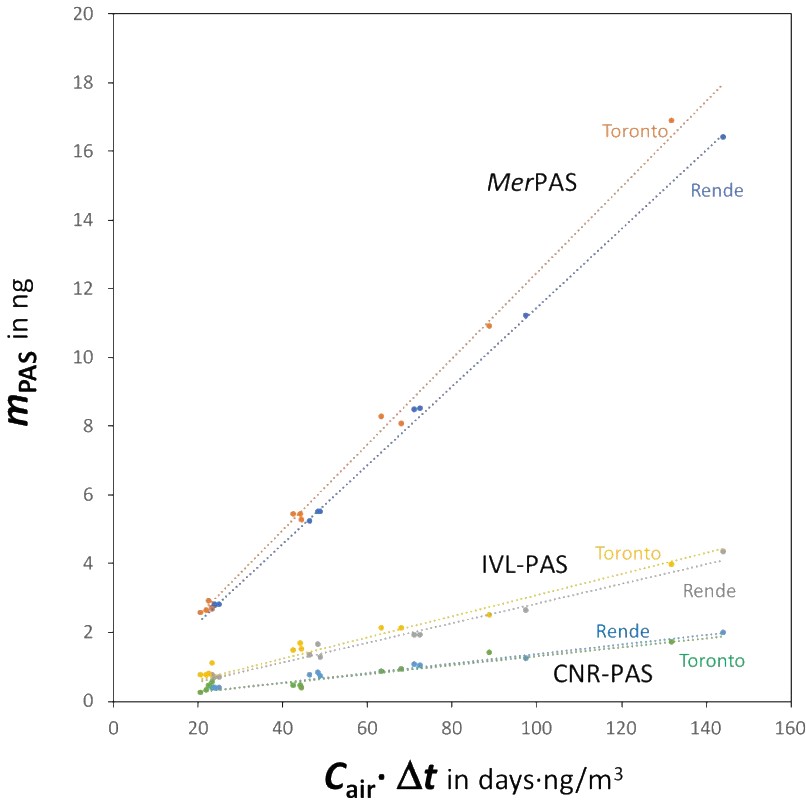

**Figure 5** Plot of the blank-corrected amount of Hg quantified in three types a passive air sampler deployed in Rende or
Toronto against the product of the deployment time of a sampler $\Delta t$ and the average air concentration during the
465       deployment of that sampler $C_{air}$, as determined independently by an Tekran active sampling system.

All uptake curves are linear with high $r^2$ values and small relative standard errors of the slope of the linear regressions

(1 % for *Mer*PAS®, 2 to 3 % for IVL-PAS, 2 to 5 % for CNR-PAS). A graphical inspection of Fig. 5 also confirms that forcing

the regression through the origin was justified, indicating that the blank correction was largely effective for all three samplers.





Consistent with what should be expected from the sampler performance at the two locations, the regressions are generally

better for the Rende than for the Toronto deployments. Overall, it is clear that all three samplers perform as true linear uptake

samplers at both locations over a three-month period.

**Table 3** Results of the linear regressions displayed in Fig. 5. The slope of the regression line corresponds to the sampling rate of a passive air sampler.

| Sampler | Location | *SR* (m$^3$ day$^{-1}$) | Relative SE (%) | *r²* | *a priori* SR (m$^3$ day$^{-1}$) | SR difference (%) |
|---------|----------|-------------------------|-----------------|------|----------------------------------|-------------------|
| *Mer*PAS® | Rende | 0.1144 ± 0.0006 | 0.5 | 0.9997 | 0.111 | +3 |
|  | Toronto | 0.1247 ± 0.0014 | 1.1 | 0.9988 | 0.111 | +12 |
| IVL-PAS | Rende | 0.0284 ± 0.0006 | 2.2 | 0.9951 | 0.03 | -5 |
|  | Toronto | 0.0309 ± 0.0009 | 3.0 | 0.9911 | 0.03 | +3 |
| CNR-PAS | Rende | 0.0139 ± 0.0003 | 2.2 | 0.9951 | 0.0147 | -6 |
|  | Toronto | 0.0131 ± 0.0007 | 5.5 | 0.9710 | 0.0147 | -11 |

475          Table 3 also compares the site- and deployment specific *SR*s obtained from the regressions with the generic *a priori*

ones that were used in the calculation of the volumetric air concentrations from the PASs. Deviations between these *SR*s should

be roughly similar to the bias of the PAS-derived air concentrations, reported in Table 2. In the case of the *Mer*PAS®, they are

indeed very similar (+2.8 % vs. +3.1 % at Rende, +10.2 vs. +12.3 % at Toronto). In the case of the IVL-PAS (-0.5 % vs. -5.4

% at Rende, +17.0 % vs. +3.0 % at Toronto) and CNR-PAS (-1.4 % vs. -5.7 % at Rende, -8.8 % vs. -10.7 % at Toronto) they

are less similar, although the direction of bias is the same. The deviations are not exactly the same, because the longer

deployments have a stronger impact on the slope of the lines in Fig. 5 than shorter ones, whereas when the average bias given

in Table 2 was calculated, each sample counted the same, irrespective of deployment length.

         We can also compare the relative size of the fitted *SR*s at the two locations. Interestingly, for both *Mer*PAS® and IVL-

PAS the *SR* was 9 % higher in Toronto than in Rende. The *SR* of the CNR sampler shows an opposite behavior, being 5 %

lower in Toronto than in Rende. Meteorological factors can be responsible for differences in *SR* between deployment at

different sites or different time period. In particular, an increase in the molecular diffusivity of Hg in air with temperatures can lead to a higher $SR$ at higher temperatures, whereas an increase in wind speed can reduce the thickness of the stagnant air boundary layer surrounding a PAS's diffusive barrier and therefore also lead to a higher $SR$ (McLagan et al., 2017). Toronto was much colder than Rende during the study period (average during the 12 weeks deployment period of 1 °C and 12 °C,

respectively), which would be consistent with a lower $SR$ in Toronto as was observed for the CNR-PAS. On the other hand, wind speeds in Toronto were approximately double those in Rende (average of 2.4 m s$^{-1}$ and 1.2 m s$^{-1}$, respectively), which would be consistent with higher $SR$ in Toronto as was observed by IVL-PAS and *Mer*PAS®.

### 3.2.5 Reasons for the different performance in Rende and Toronto

Generally, the three PASs performed better in the deployments in Rende than in those in Toronto. This is most

apparent in the assessment of accuracy (Figs. 3 and 4). However, this did not apply to all performance indicators. For example, the magnitude and variability in field blanks was comparable between the two sites for the *Mer*PAS® and IVL-PAS, while the CNR-PAS had much more variable field blank contamination in Toronto than in Rende (Fig. 1). Also, replicate precision (prior to blank correction) was very similar at the two sites (Fig. 2); in the case of the IVL-PAS, the replicate precision was in fact better in Toronto. This suggests that operator handling is unlikely to be responsible for the differences in performance at the

two sites.

A major difference between the two sites is the harshness of the weather conditions during the deployment period, which comprised the three months of February to April 2019. Winter and early spring in Toronto can be very cold, experience large temperature fluctuations over short time periods as well as precipitation in different forms (snow, freezing rain, sleet, rain). As was discussed in the preceding section, temperature and wind speed can influence the rate of diffusion to the passive

sampling sorbent, causing variability in the $SR$s. It is also conceivable that during inclement weather, hoarfrost forms on the surfaces of the diffusive barriers or blowing snow could cake up on the samplers, potentially impeding the path of Hg to the sorbent. However, it will often not be possible to attribute discrepancies to weather conditions, for example when deviations occur in opposite direction during overlapping deployments (e.g. the third 2-week and the second 4-week deployments overlap, yet the CNR-PAS shows positive bias in the former and negative bias in the latter).

Another possible source of the discrepancies between Tekran and PAS concentrations in Toronto (Fig. 3) is the higher

fraction of missing/rejected data from the Tekran operating in Toronto. Whereas in Rende, generally 98.9 % was covered by

valid Tekran data, the data coverage for the PAS deployment reached 82.5 % for the Tekran "5037" units providing the

reference value in Toronto. The reason for the lower percentage in Toronto is a sampling method that relies on daily calibrations

(2.4 % daily loss of coverage) and hourly (8.3 % daily loss of coverage) spikes; these alone already account for a 10.7 % per

day loss of coverage, yet improve confidence in the data. The distribution of the standard addition spikes throughout the day

however means they are unlikely to result in any bias of the results. The remainder was due to regular maintenance, and a

power outage. A different, but equally valid, sampling method in Rende ran calibration every 3 days and no spikes. For

individual deployments data coverage ranged as low as 66 % for the fourth 2-week deployment in Toronto. However, the

discrepancy between PASs and Tekran are not unusually large during that deployment period (Fig. 3).

A final difference between the two study locations is the occurrence of several short spikes of elevated GEM

concentrations in Toronto. If these had been caused by a local source in immediate vicinity of the sampling site, it is

conceivable that spatial GEM concentration gradients may have been present within the assembly of PASs and Tekran inlets.

However, no relationship between the occurrence of such spikes and the discrepancies in PAS results is apparent. In any case,

it is more likely the spikes were caused by sources sufficiently far from the sampling site to not result in concentration gradients

on the scale of a few meters.

## 4. CONCLUSIONS AND RECOMMENDATIONS

Table 4 compiles the key performance indicators for the three passive air samplers. In contrast to most of the sections

above, this table provides the average of all values obtained from the Rende and Toronto deployments. This compilation

reveals that the *Mer*PAS® is currently the best performing PAS among the three, having the lowest LODs, the highest precision

and the best accuracy when judged based on the discrepancy from the Tekran system. An important reason for this better

performance is the larger size and radial diffusion configuration of the *Mer*PAS®, which leads to much higher *SR*s than for the

other samplers which are axial diffusion samplers and also much smaller in size. A higher *SR* means that the amount of Hg

taken up in a *Mer*PAS® during a deployment is much higher relative to the blank contamination level than it is for the other





two samplers, which have very similar blank contamination levels. That inconsistent and relatively high blank contamination

levels could hamper the performance of a PAS is evident from the comparison of performance at the two sites. Higher and

more variable blank levels of the CNR-PAS in Toronto translate into much higher LODs (Table 2) and much poorer precision

after blank correction than in Rende. Incidentally, this also highlights pathways for improvement, namely either a reduction in

the magnitude and variability of blank contamination and/or a change in the sampler size and configuration that increases the

*SR*. A promising result of this study is that the *SR*s of the CNR-PAS at the two locations are more similar than for the other

two PAS, which may hint at a *SR* that has a relatively small dependence on meteorological factors. However, this will still

need to be confirmed by calibrating the sampler under a wide variety of meteorological circumstances.

Table 4 also shows that IVL-PAS and CNR-PAS are remarkably similar in their performance characteristics with

very similar LODs and replicate precision. While the average bias of the CNR-PAS overall is very small, this is largely because

fairly large discrepancies occur in either direction and therefore cancel each other out. Overall, the IVL-PAS derived air

concentrations agree better with the Tekran derived data than those of the CNR-PAS (12.5 vs. 19.1 %).

**Table 4** Summary of the key metrics describing the performance of the three passive air samplers for Hg as determined in this study.

| | *Mer*PAS® | IVL-PAS | CNR-PAS |
|---|---|---|---|
| MDL (ng) | 0.16 | 0.25 | 0.13 |
| LOD (2 weeks) (ng m$^{-3}$) | 0.10 | 0.59 | 0.65 |
| LOQ (2 weeks) (ng m$^{-3}$) | 0.34 | 1.98 | 2.16 |
| LOD (12 weeks) (ng m$^{-3}$) | 0.02 | 0.10 | 0.11 |
| LOQ (12 weeks) (ng m$^{-3}$) | 0.06 | 0.33 | 0.36 |
| Replicate precision (before blank correction) (%) | 3 | 9 | 7 |
| Replicate precision (after blank correction) (%) | 4 | 15 | 14 |
| Concentration bias (relative to Tekran), n = 22 (%) | +6.5 | +8.2 | -5.1 |
| Absolute discrepancy (relative to Tekran), n = 22 (%) | 6.5 | 12.5 | 15.5 |
| Linear uptake over 12 weeks | Yes | Yes | Yes |

The *Mer*PAS®-derived air concentrations were on average 6.5 % higher than the Tekran-derived values, and this positive bias was evident at both deployment locations, albeit more pronounced in Toronto. Discrepancies with Tekran data

of similar size have previously been reported for the sampler by (McLagan et al., 2016a), on which the *Mer*PAS® is based. For example, (McLagan et al., 2018) reported an average discrepancy of ~9 %. Nevertheless, the results presented here indicate that the generic *SR* of 0.111 m$^3$/day applied here may benefit from further refinement and should possibly be somewhat higher.

*Authors contribution*

**Attilio Naccarato:** Conceptualization, Methodology, Data Curation, Writing - Original Draft, Writing - Review & Editing,

Supervision; **Antonella Tassone, Maria Martino, Sacha Moretti:** Investigation (Rende field deployments and TEKRAN measurements), Formal analysis, Data Curation, Writing - Original Draft, Writing - Review & Editing; **Antonella Macagnano, Emiliano Zampetti, Paolo Papa, Joshua Avossa:** Preparation and analysis of CNR-PAS; **Nicola Pirrone, Francesca Sprovieri:** Funding acquisition; **Michelle Nerentorp, John Munthe, Ingvar Wängberg:** Preparation, analysis and data evaluation of IVL PAS, Writing - Review and editing; **Geoff W. Stupple**: Investigation (Toronto field deployments and

TEKRAN measurements), Writing - Review & Editing; **Carl P. J. Mitchell**, **Alexandra Steffen**, **Eric M. Prestbo**: Methodology, Writing - Review & Editing; **Adam R. Martin**: Formal Analysis (Statistical analysis), Writing - Review & Editing; **Diana Babi**: Investigation (Analysis of MerPAS); **Frank Wania**: Conceptualization, Methodology, Data Curation, Visualization, Writing - Original Draft

*Acknowledgement*

565       We are grateful to David Gay for compiling the data generated by the different participants and making them available after they had all been received, and to Emily Alvarez for help with sampler deployments in Toronto. The Ontario Ministry of Environment, Conservation and Parks is acknowledged for site access and meteorological data. The work in Toronto was supported by a Discovery Grant of the Natural Sciences and Engineering Research Council of Canada to F.W. and a grant and contribution agreement from Environment and Climate Change Canada (fund #GCXE19S042) with C.P.J.M. The work in

Rende was funded by the European Commission—H2020, the ERA-PLANET program (www.era-planet.eu; contract no. 689443) within the IGOSP project (www.igosp.eu). The study was also supported by the MercOx project (EMPIR EURAMET,



project number: 16ENV01). IVL would like to acknowledge the financial support from the EU ERA-Net Cofund program

ERA Planet – iGOSP, Grant number 689443, and the Foundation for the Swedish Environmental Research Institute (SIVL).

*Conflict of Interest Statement*

575         The author declare the following competing financial interest(s): Tekran Instruments Corp. pays some licensing fees

to the University of Toronto related to the *Mer*PAS®, which are in part being distributed to co-authors C. P. J. M. and F. W..

E.M.P. and D. B. are employees of Tekran Instruments Corp., which sells the *Mer*PAS® commercially.



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
