# Peer review of "A field intercomparison of three passive air samplers for gaseous mercury in ambient air"

_Atmospheric Measurement Techniques, 2020_

## Referee Comment (RC1) · Anonymous Referee #1 · 2 Jan 2021

This papers describes tests of 3 different passive samplers for gaseous elemental mercury. These samplers are typically deployed for 8 to 12 weeks. I disagree with the first sentence in the abstract. They will tell us little to nothing. These samplers are only useful in contaminated areas and are not appropriate for monitoring. Gradients in air concentrations between the Northern and Southern hemisphere have been well documented. They are not highly variable. Thus, as for these helping meet assessment needs these samplers, because of the long sampling time will tell us nothing about the "presence and movement of mercury and mercury compounds in the environment." We know gaseous elemental mercury is ubiquitous in the air. These sampling systems will tell us little about transport and deposition or mercury compounds. The authors need to be honest about these methods and what information will be gained, for in my mind

this is little.

Abstract and throughout the paper what has been done should be described in the past tense. Line 28 remove both Line 46 remove highly Line 84 remove for the first time Lines 95, 189, 344, and 534 remove ", which" and replace with "that" Line 221 put a comma before because Comment on sampling rate calculation- this will depend on wind and possibly temperature and RH, and I am not sure how this is accounted for since this will vary by each location Line 264 you should put the value for each sampler and the standard deviation Paragraph that starts line 265. Were values measured by the two Tekrans the same? Typically they are systematically different. As you can see regarding the discussion of the data the Tekrans measured the variability while the passive samplers only measure the average and tell us little about the presence and movement of mercury or the compounds. Also please put the standard deviations in for each sampler. Are the values statistically significantly different between Italy and Canada using these samplers? I would expect Toronto to be a bit higher due to the fact it is a big city. Were the blank corrections subtracted for each sampling interval or just across the whole time? The former is the best way to do this and it is not clear in the text how this was done. Line 385 –how do you know which Tekran concentrations were right? Discussion regarding sampling rate. This demonstrates how this will be variable depending on conditions and they need a way to adjust for this. If you are at a remote location and are not collecting meteorological data how can you calculate the SR? If sites have different meteorological conditions and you have no Tekran system how will you know how accurate your values are? Given the low concentrations what will this information tell us? Basically nothing useful. Again. . . these passive sampling systems did not capture spikes. So what is the utility of these methods?

The authors should also look at the literature and discuss other passive sampling systems that have been tested for gaseous elemental mercury.

---

## Referee Comment (RC2) · Anonymous Referee #4 · 21 Jan 2021

The paper is well written and as the title suggests relates to an inter-comparison of 3 passive samplers for gaseous mercury in ambient air. Throughout the paper the analyte is referred to as gaseous mercury which by definition would normally include both elemental Hg (GEM) and reactive gaseous Hg species (RGM). There are no validation studies to my knowledge that report that passive samplers will determine RGM and therefore it would be more appropriate to refer the analyte as GEM and also provide additional discussion on this point.

The use of passive samplers has limited value to fully understanding the cycling of Hg in the atmosphere as they only provide an average concentration over the deployment period. Generally, the legislation for ambient air only considers annual averages as representative of long term exposure. In this respect, passive samplers add additional

value to monitoring networks especially if they are deployed in remote locations. There is no discussion in the paper regarding manual sampling methods that use gold traps with a vacuum pump which is more comparable to passive samplers than an automated ambient air monitor such as the Tekran 2537. Manually sampling does require a power supply to operate the vacuum pump but they can also be used with rechargeable battery packs which for several weeks and they also offer total gaseous mercury (TGM) and much larger sample volumes. There are several aspects of the analytical performance evaluation of the passive samplers in this paper that are questionable and therefore need revising.

The detection limit calculations are based solely on the variability of the field blanks and currently ignore the actual blank values which are significant in relation to the mass of mercury collected on the passive samplers studied. Reporting detection limits less than the blank valve is questionable as it is not possible to quantify a mass of mercury less than the blank. The authors should recalculate the detection limits according the following IUPAC expression (LOD = Blank + 3$\sigma$n-1). This is far more appropriate way to report LODs for techniques that employ pre-concentration.

It is well known that sampling rates for passive samplers are affected by temperature, pressure, humidity and wind speeds. In this study these parameters are ignored eventhough the metrological conditions at the sampling location have been measured and are available. The automated Tekran analyzers are based on a standardized volume measurement and the passive samplers are not. If the authors ignore the metrological conditions then the comparison is not valid. At both sites the metrological conditions were highly variable and in addition to this no pressure correction has been made for barometric pressure and elevation on each site. The impact of metrological conditions on sampling rates is discussed later on in the paper as an explanation for the higher variability at the Canadian site so why was the volumetric correction not applied? I would prefer that all passive sampler results are reported using a standardized volume based on the average metrological conditions on site for the deployment period. It

would also be good to know how the uncertainty of the sampling rate is affected by the metrological conditions. Whilst the overall correction might be small it is still important when performing and report the comparison. Humidity is unlikely to have an impact. Wind speed could have an impact but since the samplers are housed in a shelter then one could argue that this could also be ignored. The correction should therefore focus on temperature and pressure. for the Accuracy assessment was performed by comparing the result the Tekran analyzers operating on each site. The paper explains that two Tekran instruments were operated but it seems that results from only one analyzer was used for the comparison? Surely, the comparison should be done on the average of the two Tekran results? If there are periods of known downtime for one of the two analyzers or technical justifications for ignoring a period of time then that is understandable but to ignore one analyzer and assume that the second analyzer is accurate does not seem correct. It seems that the authors have adopted a strategy of running analyzers in parallel but only using the second analyzer as reassurance them that the other analyzer was performing well. It would be nice to see the trend for both analyzers especially when so called spike events are experienced. Other concerns relate to the online analyzer reporting GEM even though the description of the analyzer suggests that TGM was determined because sampling was performed directly onto gold traps? The authors need to focus on what they are actually measuring with each technique applied. As mentioned above the Tekran measurements are based on a standardized volume whereas the passive samplers are not. It would be acceptable in the paper to discuss this in more detail and refer to comparability rather than accuracy. Having a passive sampler offering good agreement to the automated measurement is a very reassuring but to define the accuracy on this basis is not acceptable.

Overall this is very interesting study and the publication will be interest to many readers conducting mercury research in the field of atmospheric measurements. If passive samplers are deployed at more sites in the future then it is important to standardize the sample rate volumes especially when performing comparisons with automatic measurement systems.

---

## Referee Comment (RC3) · Anonymous Referee #2 · 23 Jan 2021

I found the study to be an informative one, concerning the comparison of three different passive GEM samplers. The comparison with Tekrans and on two continents was also informative and highlights some of the limitations but also benefits of deploying these samplers. The authors provide some insightful analysis of the discrepancies they observed.

---

## Referee Comment (RC4) · Anonymous Referee #3 · 1 Feb 2021

This paper was well written in general, as there are English first language authors, although there is a small section that maybe was added and not checked by them were the language needs revising.

The use of passive samplers is very important as many places are inaccessible Tekran type instruments. It would be useful for the reader not know wht sort of corrections are required based on temperature, wind speed, and air pressure, so they can evaluate their usefulness for their site. Otherwise a useful comparison of a very necessary low cost low power mercury monitoring system.

---

## Author Comment (AC1) · 15 Mar 2021

* * *
We appreciate the positive appraisal of our study

---

## Author Comment (AC2) · 15 Mar 2021

**Response to Comments by Anonymous Referee #3**

**This paper was well written in general, as there are English first language authors, although there is a small section that maybe was added and not checked by them were the language needs revising.**

We would appreciate if the reviewer could identify the small section where the language needs revising. A co-author whose first language is English will proof-read the final version of the text.

**The use of passive samplers is very important as many places are inaccessible to Tekran type instruments. It would be useful for the reader not know what sort of corrections are required based on temperature, wind speed, and air pressure, so they can evaluate their usefulness for their site. Otherwise a useful comparison of a very necessary low-cost low power mercury monitoring system.**

We appreciate the overall positive assessment of the merits of our study.

In the current evaluation of the three passive air samplers, we have chosen to use sampling rates that are assumed invariant with respect to meteorological conditions. In other words, the performance metrics involving a comparison with the Tekran-derived air concentrations (e.g. Table 2, Figure 3) apply to a situation when a PAS is used with a constant sampling rate. Note, however, that many of the performance metrics (LOD, LOQ, precision) are independent of the chosen sampling rate.

One reason for assuming a constant sampling rate in the current comparison is that the dependence of the sampling rate on temperature and wind speed has so far only been quantified for one of the samplers, i.e. the sampler developed by the University of Toronto on which the *Mer*PAS® is based (McLagan et al., 2018a). On lines 115 to 116, we had written: "Results for the *Mer*PAS® with temperature-adjusted sampling rates, that had also been submitted were disregarded, as they were the only set of results seeking to take into account this effect." As stated in response to comments by reviewer #1, the variability of the sampling rate of the IVL-PAS and CNR-PAS with meteorological parameters still needs to be fully investigated.

We should also add, that the evidence of the value of adjusting sampling rates to the wind speeds and temperatures prevailing during a PAS's deployment is still somewhat mixed:

- McLagan et al. (2018b) concluded in their evaluation of the MerPAS at numerous sites around the world: "Adjusting the sampling rate to deployment-specific temperatures and wind speeds does not decrease the difference in active–passive concentration further, but reduces its variability by leading to better agreement in Hg concentrations measured at sites with very high and very low temperatures and very high wind speeds."

- On line 117 of the current manuscript, we had indicated that: "Temperature adjustment did not improve the accuracy of the *Mer*PAS® results."

We suggest that more evaluations of the PAS performance at sites with extremes in temperature and wind speed will be required to judge the merit of adjusting the sampling rate for deployment-specific conditions.

**References**

McLagan, D.S., B. Abdul Hussain, H. Huang, Y.D. Lei, F. Wania, C.P.J. Mitchell. Identifying and evaluating urban mercury emission sources through passive sampler-based mapping of atmospheric concentrations. *Environ. Res. Lett.* **2018a**, *13*, 074008.

McLagan, D.S., C.P.J. Mitchell, A. Steffen, H. Hung, C. Shin, G.W. Stupple, M.L. Olson, W.T. Luke, P. Kelley, D. Howard, G.C. Edwards, P.F. Nelson, H. Xiao, G.-R. Sheu, A. Dreyer, H. Huang, B. Abdul Hussain, Y. D. Lei, I. Tavshunsky, F. Wania. Global evaluation and calibration of a passive air sampler for gaseous mercury. *Atmos. Chem. Phys.* **2018b**, *18*, 5905-5919.

---

## Author Comment (AC3) · 15 Mar 2021

**Response to Comments by Anonymous Referee #1**

This paper describes tests of 3 different passive samplers for gaseous elemental mercury. These samplers are typically deployed for 8 to 12 weeks. I disagree with the first sentence in the abstract. They will tell us little to nothing. These samplers are only useful in contaminated areas and are not appropriate for monitoring. Gradients in air concentrations between the Northern and Southern hemisphere have been well documented. They are not highly variable. Thus, as for these helping meet assessment needs these samplers, because of the long sampling time will tell us nothing about the "presence and movement of mercury and mercury compounds in the environment." We know gaseous elemental mercury is ubiquitous in the air. These sampling systems will tell us little about transport and deposition or mercury compounds. The authors need to be honest about these methods and what information will be gained, for in my mind this is little.

Considering that most of the authors of the paper expended considerable effort, time, and resources on the development of passive air samplers (PASs) for gaseous mercury, it probably comes not as a surprise that we assess the usefulness of those samplers profoundly differently from the reviewer and do not share the opinion that "they will tell us little to nothing". Reviewing the merits of PASs for gaseous elemental mercury (GEM), McLagan et al. (2016) identified primarily three areas where such samplers complement existing active sampling systems: "(1) long-term monitoring of atmospheric GEM levels in remote regions and developing countries, (2) atmospheric mercury source identification and characterization through finely resolved spatial mapping, and (3) the recording of personal exposure to GEM." Since the writing of this review five years ago, PASs for GEM have already proven their usefulness in all three of these application areas.

**Re (1): Long-term monitoring of atmospheric GEM levels in remote regions and in developing countries.** There are a number of site categories where measurements of GEM with existing active sampling techniques are logistically extremely challenging, prohibitively expensive or outright impossible. The reasons include:

- An electrical power supply is non-existent, intermittent or unreliable.
- Site access is difficult or only possible sporadically (e.g. research stations in remote areas, which are only serviced by vessels once or a few times in a year).
- The likelihood of loss of, or damage to, sampling equipment by natural forces or human intervention is high (e.g. sampling on volcanoes (Si et al., 2020), on sea ice, or in locations with illicit mercury use (Snow et al., 2021b)).

The simplicity, low cost, and limited site requirements (with respect to electrical power, etc.) often make it feasible to use PASs under such circumstances and will greatly expand the number of sites where GEM concentrations can be recorded.

In the map below, taken from the Global Mercury Assessment 2018 (AMAP/Un Environment 2019), the global distribution of model ensemble median gaseous elemental mercury concentration in surface air for 2015 are shown and compared with observed values. The places where GEM has been measured in the atmosphere (shown with circles using the same colour scale) show glaring spatial bias. There are large swathes of the global atmosphere where no such measurements have been conducted. Passive air samplers are poised to fill many of those data gaps.

We further point out, that PASs are not just suited to record the long-term average concentration of GEM in the atmosphere, but can also be used to record the isotopic composition of that mercury (Szponar et al., 2020).

**Re (2): Atmospheric mercury source identification and characterization through finely resolved spatial mapping.** One of the primary capabilities of PASs is the ability to measure simultaneously at numerous locations. This is again feasible, because of the low cost and limited site requirements relative to those of most active air sampling techniques. As a result, PASs have for the first time allowed for the mapping of the spatial concentration variability of GEM. While there are portable active sampling systems that have been used for mapping before, they typically make sequential short-term measurements at multiple locations and therefore cannot clearly discriminate temporal from spatial concentration variability. PASs for GEM have already been used to map spatial GEM concentration variability within a large urban area (McLagan et al. 2018a), in the vicinity of an abandoned mercury mine (McLagan et al., 2019), in geothermally active areas (Si et al., 2020), in artisanal gold-mining communities (Snow et al., 2021b) and in occupational environments (Snow et al., 2021b).

Monitoring for GEM does not have to be focussed on atmospheric background concentrations, but in fact is often called for in "contaminated areas". The objective of such monitoring then could be (i) the identification or confirmation of atmospheric GEM emission sources (McLagan et al., 2018a), (ii) the quantification of atmospheric GEM emissions, whether they are anthropogenic (McLagan et al., 2019) or geogenic (Si et al., 2020), or (iii) the characterisation of human inhalation exposure (Snow et al. 2021b).

**Re (3): Recording of personal exposure to GEM.** If worn in the breathing zone of an individual, PASs can record the personal inhalation exposure to GEM (Snow et al., 2021a). This has been demonstrated e.g. for the exposure of the acute and chronic exposure of people in artisanal and small-scale gold mining (Snow et al. 2021b, Black et al., 2017, de Barros Santos et al., 2017).

In summary, the reviewer appears to have a very limited imagination as to what "monitoring of atmospheric mercury worldwide" could possibly comprise. We are looking forward to proving the reviewer wrong in predicting that little of merit will be gained from the type of sampler evaluated in our study.

**Abstract and throughout the paper what has been done should be described in the past tense.**

**Line 28 remove both**

Line 46 remove highly

These are largely suggestions on writing style rather than required corrections in grammar and thus we do not see the need for adopting these changes. Just as an example, every compound is "toxic" depending on dose, so to characterize mercury as "highly toxic" seems hardly inappropriate.

**Line 84 remove for the first time**

We believe to have presented in the manuscript the first blind inter-comparison of passive air samplers for mercury and therefore this statement to be factual. If the reviewer can provide a reference to other such studies and we will reference them and alter the text.

**Lines 95, 189, 344, and 534 remove ", which" and replace with "that"**

All four of these clauses starting with "which" are not defining clauses and therefore the use of "which" is grammatically correct. As it is possible to remove the clause without destroying the meaning of the sentence, the clause is non-essential and "which" can be used. We, therefore, think that no change to the grammar for those sentences is required.

**Line 221 put a comma before because**

We could add a comma during the revision.

**Comment on sampling rate calculation- this will depend on wind and possibly temperature and RH, and I am not sure how this is accounted for since this will vary by each location**

On line 219 we indicated that constant sampling rates for all the samplers were applied. We could include the phrase "these rates were not adjusted for deployment-specific meteorological conditions" in a revised version.

Furthermore, on lines 115 to 117, we had written: "Results for the *Mer*PAS® with temperatureadjusted sampling rates, that had also been submitted were disregarded, as they were the only set of results seeking to take into account this effect. Temperature adjustment did not improve the accuracy of the *Mer*PAS® results."

Lines 485-492 include a discussion of the possibility that the sampling rates are affected by wind speed and temperature and whether those influences could explain the observed discrepancies between active and passive sampling results.

Finally, we also had written on line 539: "A promising result of this study is that the SRs of the CNR-PAS at the two locations are more similar than for the other two PAS, which may hint at an SR that has a relatively small dependence on meteorological factors."

**Line 264 you should put the value for each sampler and the standard deviation**

The concentration values given in this line refer to the measurements obtained by active sampling. On lines 263 and 271, we already report the standard deviation of the actively recorded air concentrations.

Paragraph that starts line 265. Were values measured by the two Tekrans the same? Typically they are systematically different. This information is provided on lines 384-389: "the "0075" instrument yielded values averaged over the deployment periods that were consistently lower than those measured by the "5037" instrument that was chosen as the reference. This bias was on average 3.2 % for the 11 sampling periods and ranged from a low of 1.0 % for the third 2-week period to a high of 6.5 % for the last 4-week period." Although concentrations were consistently lower on the "0075", it was not a systematic difference that could be corrected and this resulted in data not meeting QC criteria. This is addressed on lines 267-269: "The secondary co-located 2537a analyzer experienced an 8% shift in the mass-flow meter calibration during the study. Since it was not possible to determine when the shift occurred, data from this analyzer were not used for comparison with the PAS."

**As you can see regarding the discussion of the data the Tekrans measured the variability while the passive samplers only measure the average and tell us little about the presence and movement of mercury or the compounds.**

While it is often useful to know the variability of the GEM concentration at fine temporal resolution, there are many instances where the average of that concentration is of primary interest. In fact, when analyzing spatial and temporal variation in GEM concentrations, data measured at high resolution are regularly aggregated into monthly, seasonal and annual concentration bins (see e.g. Cole et al., 2014, Weiss-Penzias et al., 2016), indicating that those time periods of analysis convey valuable information on the presence and movement of mercury in the environment. These time periods are very well served with a PAS.

**Also please put the standard deviations in for each sampler. Are the values statistically significantly different between Italy and Canada using these samplers? I would expect Toronto to be a bit higher due to the fact it is a big city.**

The average concentration values as measured by each PAS were reported in Table S8 as mean  $\pm$  sd. We do not see any merit in testing whether those two concentrations are statistically significantly different between the two locations. As is appropriate for a study that is comparing different sampling techniques, statistical analysis is focussed on the differences between samplers and not between locations. The average GEM concentration with standard deviation as measured by the Tekran systems over the twelve weeks of the entire study was 1.72  $\pm$  0.25 ng m-3 in Italy (as given on line 262) and 1.57  $\pm$  0.45 ng/m-3 in Canada (as provided on line 270), i.e. the opposite of the reviewer's expectations. Note that the Toronto location is in the north end of the city next to a large park and not in downtown metropolitan Toronto, while in Rende the sampling was carried out at a site potentially affected by vehicular traffic from a highway, a nearby urban area, and also by the presence of a small manufacturing area a few kilometers away.

**Were the blank corrections subtracted for each sampling interval or just across the whole time? The former is the best way to do this and it is not clear in the text how this was done.**

The amount quantified in each individual sampler was corrected for the average blank contamination obtained at a sampling site (see lines 115 and 218). To make this clearer, we can add the following information to section 2.7 in a revised version: "Because field blank levels were

not statistically significantly different between different deployment periods, but were different between Rende and Toronto, the average field blank contamination at one location was used for blank correction."

**Line 385 -how do you know which Tekran concentrations were right?**

There is no way of knowing which Tekran system measures the "right" concentration. However, the system labelled "5037" met the QC criteria, whereas the system labelled "0075" "experienced an 8 % shift in the mass-flow meter calibration during the study" and "since it was not possible to determine when the shift occurred" (line 268), we decided to use the system labelled "5037" for comparison with the passive sampling results. When there is a known issue with an instrument but we are not sure when the problem started to occur, we err on the side of caution and use the data in which we have the most confidence based on our QC protocols.

**Discussion regarding sampling rate. This demonstrates how this will be variable depending on conditions and they need a way to adjust for this.**

The reviewer is correct that the sampling rates can be somewhat influenced by local conditions during a deployment period, most importantly temperature and wind speed. In the case of the *Mer*PAS®, the dependence of the sampling rate on those conditions has been quantified in extensive laboratory experiments (McLagan et al. 2017) and a way to adjust for this influence during field deployment has also been presented (McLagan et al., 2018b). In the case of the CNR-PAS, the influence of meteorological parameters has not yet been fully investigated. However, laboratory experiments on the performance of the sorbent membrane at different temperature and humidity conditions have been conducted (Macagnano et al., 2018, Avossa et al., 2020).

**If you are at a remote location and are not collecting meteorological data how can you calculate the SR?**

As previously explained, the dependence of the sampling rates on meteorological variables is small. Because only the average conditions during the entire deployment period are relevant, it will often be possible to use temperature and wind speed conditions estimated from climate normal. An example of how this can be done for sampling sites around the world, see Herkert et al. (2018).

**If sites have different meteorological conditions and you have no Tekran system how will you know how accurate your values are?**

It is possible to assess the accuracy of a PAS-derived air concentration based on evaluation studies such as the one described in the current paper. In the case of the *Mer*PAS®, an earlier evaluation study assessed sampler accuracy at numerous sampling sites across the world with a wide range in climatic conditions (McLagan et al., 2018b). As for the more recently developed CNR-PASs, the effect of meteorological conditions is still under investigation. A study regarding the performance at different temperatures and humidities has recently been published (Avossa et al., 2020). The current paper about the accuracy of the sampler will contribute to the future evaluation of the CNR-PAS's performance.

**Given the low concentrations what will this information tell us? Basically nothing useful. Again. . . these passive sampling systems did not capture spikes. So what is the utility of these methods?**

We refer to the detailed response to similar sentiments expressed by the reviewer above. Again, while there can be situations when "spikes" in concentrations are important, there are also

numerous instances where temporally averaged concentrations (even if they happen to be low) are perfectly adequate for meeting monitoring objectives.

**The authors should also look at the literature and discuss other passive sampling systems that have been tested for gaseous elemental mercury.**

Some of us have reviewed earlier passive sampling systems for GEM in McLagan et al. (2016) and concluded that "none of the GEM PASs developed to date achieve levels of accuracy and precision sufficient for the reliable determination of background concentrations over extended deployments". Incidentally, this judgement was part of the driving force for developing the PASs comparatively assessed in this study. We don't see a need to repeat a discussion of these earlier systems here.

**References**

AMAP/UN Environment. Technical Background Report for the Global Mercury Assessment 2018. Arctic Monitoring and Assessment Programme, Oslo, Norway/UN Environment Programme, Chemicals and Health Branch, Geneva, Switzerland, **2019**. viii + 426 pp including E-Annexes.

Avossa, J., F. De Cesare, P. Papa, E. Zampetti, A. Bearzotti, M. Marelli, N. Pirrone, A. Macagnano. Characteristics and performances of a nanostructured material for passive samplers of gaseous Hg. *Sensors*, **2020**, *20*, 6021.

Black, P., M. Richard, R. Rossin, K. Telmer. Assessing occupational mercury exposures and behaviours of artisanal and small-scale gold miners in Burkina Faso using passive mercury vapour badges. *Environ. Res.* **2017**, 152, 462-469.

Cole, A. S., A. Steffen, C. S. Eckley, J. Narayan, M. Pilote, R. Tordon, J. A. Graydon, V. L. St Louis, X. Xu, B. Branfireun. A Survey of Mercury in Air and Precipitation across Canada: Patterns and Trends, Atmosphere **2014**, *5*, 635-668.

de Barros Santos, E.; P. Moher, S. Ferlin, A. H. Fostier, I. O. Mazali, K. Telmer, A. Guimarães Brolo. Proof of concept for a passive sampler for monitoring of gaseous elemental mercury in artisanal gold mining. *Sci. Rep.* **2017**, *7*, 16513.

Herkert, N. J., S. N. Spak, A. Smith, J. K. Schuster, T. Harner, A. Martinez, K. C. Hornbuckle. Calibration and evaluation of PUF-PAS sampling rates across the Global Atmospheric Passive Sampling (GAPS) network. *Environ. Sci.: Processes Impacts* **2018**, 20, 210-219.

Macagnano, A., P. Papa, J. Avossa, V. Perri, M. Marelli, F. Sprovieri, E. Zampetti, F. De Cesare, A. Bearzotti, N. Pirrone: Passive sampling of gaseous elemental mercury based on a composite TiO2NP/AuNP layer. *Nanomaterials* **2018**, *8*, 798, doi:10.3390/nano8100798.

McLagan, D.S., M.E.E. Mazur, C.P.J. Mitchell, F. Wania. Passive air sampling of gaseous elemental mercury: A critical review. *Atmos. Chem. Phys.* **2016**, *16*, 3061–3076.

McLagan, D. S., C. P. J. Mitchell, H. Huang, B. Abdul Hussain, Y. D. Lei, F. Wania. The effects of meteorological parameters and diffusive barrier reuse on the sampling rate of a passive air sampler for gaseous mercury. *Atmos. Meas. Tech.* **2017**, 10, 3651-3660.

McLagan, D.S., B. Abdul Hussain, H. Huang, Y.D. Lei, F. Wania, C.P.J. Mitchell. Identifying and evaluating urban mercury emission sources through passive sampler-based mapping of atmospheric concentrations. *Environ. Res. Lett.* **2018a**, *13*, 074008.

McLagan, D.S., C.P.J. Mitchell, A. Steffen, H. Hung, C. Shin, G.W. Stupple, M.L. Olson, W.T. Luke, P. Kelley, D. Howard, G.C. Edwards, P.F. Nelson, H. Xiao, G.-R. Sheu, A. Dreyer, H. Huang, B. Abdul Hussain, Y. D. Lei, I. Tavshunsky, F. Wania. Global evaluation and calibration of a passive air sampler for gaseous mercury. *Atmos. Chem. Phys.* **2018b**, *18*, 5905-5919.

McLagan, D.S., F. Monaci, H. Huang, Y.D. Lei, C.P.J. Mitchell, F. Wania. Characterization and quantification of atmospheric mercury sources using passive air samplers. *J. Geophys. Res. Atmos.* **2019**, *124*, 2351-2362.

Si, M., D. S. McLagan, A. Mazot, N. Szponar, B. Bergquist, Y. D. Lei, C. P.J. Mitchell, F. Wania. Measurement of atmospheric mercury over volcanic and fumarolic regions on the North Island of New Zealand using passive air samplers. *ACS Space Earth Chem.* **2020**, *4*, 2435-2443.

Snow, M.A., G. Darko, O. Gyamfi, E. Ansah, K. Breivik, C. Hoang, Y. D. Lei, F. Wania Characterization of inhalation exposure to gaseous elemental mercury during artisanal gold mining and e-waste recycling through combined stationary and personal passive sampling *Environ. Sci.: Processes Impacts* **2021**b, DOI: 10.1039/d0em00494d.

Snow, M.A., M. Feigis, Y. D. Lei, C. P. J. Mitchell, F. Wania. Development, characterization, and testing of a personal passive sampler for measuring inhalation exposure to gaseous elemental mercury. *Environ. Int.* **2021a**, *146*, 106264.

Szponar, N., D. S. McLagan, R. Kaplan, C. P. J. Mitchell, F. Wania, A. Steffen, G. Stupple, B. Bergquist. Isotopic characterization of atmospheric gaseous elemental mercury by passive air sampling. *Environ. Sci. Technol.* **2020**, *54*, 10533-10543.

Weiss-Penzias, P.S., D. A. Gay, M. E. Brigham, M. T. Parsons, M. S. Gustin, A. ter Schure. Trends in mercury wet deposition and mercury air concentrations across the U.S. and Canada. *Sci. Total Environ.* **2016**, *568*, 546-556.

---

## Author Comment (AC4) · 15 Mar 2021

**Response to Comments by Anonymous Referee #4**

**The paper is well written and as the title suggests relates to an inter-comparison of 3 passive samplers for gaseous mercury in ambient air. Throughout the paper the analyte is referred to as gaseous mercury which by definition would normally include both elemental Hg (GEM) and reactive gaseous Hg species (RGM). There are no validation studies to my knowledge that report that passive samplers will determine RGM and therefore it would be more appropriate to refer the analyte as GEM and also provide additional discussion on this point.**

The choice of the somewhat ambiguous term "gaseous mercury" (as opposed to "total gaseous mercury (TGM)" or "gaseous elemental mercury (GEM)") is deliberate, because not for all of the three samplers has the nature of the sampled mercury species been unequivocally established. While we agree with the reviewer that the likelihood of RGM to pass unhindered through the diffusive barriers of any of the three PASs used in this study is very small and therefore GEM is the most likely mercury species to be sampled, only for the *Mer*PAS® is there experimental evidence for this to be the case. Specifically, this paragraph is taken from the supporting information file of Si et al. (2020), which describes the use of the sampler developed by the University of Toronto, on which the *Mer*PAS® is based:

> "Previous publications presenting results using this PAS have referred to the analyte as gaseous Hg, but hypothesized that reactive gaseous oxidized mercury (GOM) is scavenged by the diffusive barrier (McLagan et al., 2016, McLagan et al., 2018a, McLagan et al., 2018b). Recent evidence has confirmed that GEM is indeed the mercury species sampled by the PAS. In an unpublished study (Stupple et al. 2019), two Tekran 2537/1130 speciation systems were run simultaneously under elevated GOM conditions during atmospheric mercury depletion events (AMDEs) at Alert in the Canadian High Arctic. After replacing the inlet of one system with the Radiello® diffusive barrier (no sorbent inside), the measured GOM dropped to near zero, while the other system continued to measure high GOM concentrations. This shows that GOM cannot pass through the diffusive barrier. Szponar et al. (2020) set up PASs with and without the Radiello® diffusive barrier during AMDEs at Alert. Isotopic analysis showed Hg isotope signatures distinctive of GOM in samples collected without the diffusive barrier, but not in samples collected with the diffusive barrier. Therefore, we are confident that GEM is an accurate description of the target analyte."

**The use of passive samplers has limited value to fully understanding the cycling of Hg in the atmosphere as they only provide an average concentration over the deployment period. Generally, the legislation for ambient air only considers annual averages as representative of long-term exposure. In this respect, passive samplers add additional value to monitoring networks especially if they are deployed in remote locations.**

First, we note that the minimum deployment period of a PAS could be quite short if atmospheric GEM concentrations are elevated. In areas affected by large GEM sources, resolution on the time scale of an hour or even shorter is feasible (Snow et al., 2021b). More importantly, as already stated in response to the comment by reviewer #1, we believe there is considerable merit to passive air sampling for GEM that goes well beyond long-term monitoring in remote locations. No single sampling/measurement approach is sufficient to fully understand the cycling of Hg. We note that neither automated ambient air monitoring by an instrument such as the Tekran 2537 can provide a full understanding of the cycling of Hg in the atmosphere as it can only provide concentrations at a very limited number of sampling sites. The key is that passive and active sampling are highly complementary and in combination are far more likely to get us the full understanding that we seek.

**There is no discussion in the paper regarding manual sampling methods that use gold traps with a vacuum pump which is more comparable to passive samplers than an automated ambient air monitor such as the Tekran 2537. Manually sampling does require a power supply to operate the vacuum pump but they can also be used with rechargeable battery packs which for several weeks and they also offer total gaseous mercury (TGM) and much larger sample volumes.**

The reviewer is correct that this type of sampler is not discussed in the paper. However, the paper does not address the question of the merits of different types of atmospheric mercury sampling techniques. It compares the performance of three different passive air samplers. While there are some potential applications of PASs for GEM that could also be accomplished with "manual sampling methods that use gold traps with a vacuum pump", many other potential PAS applications would not be amenable to that type of sampling. In the future, it may be worthwhile to pursue a similar inter-comparison exercise involving PASs and the type of sampling approach advocated by the reviewer.

**There are several aspects of the analytical performance evaluation of the passive samplers in this paper that are questionable and therefore need revising.**
**The detection limit calculations are based solely on the variability of the field blanks and currently ignore the actual blank values which are significant in relation to the mass of mercury collected on the passive samplers studied. Reporting detection limits less than the blank valve is questionable as it is not possible to quantify a mass of mercury less than the blank. The authors should recalculate the detection limits according the following IUPAC expression (LOD = Blank + 3$\sigma$n-1). This is far more appropriate way to report LODs for techniques that employ pre-concentration.**

There are different ways of calculating limits of detection and the procedure we have used is commonly applied, widely accepted and scientifically defensible. If field blanks are consistently at a certain level (with a small standard deviation) it has no impact on the ability of a method to detect an analyte, even if that field blank level may be high relative to the amount of the analyte added during sampling. It also does not imply that a mass of mercury less than the blank is being quantified. It is the mass of mercury in the field blank plus the mass of mercury added to the sorbent during sampling that is quantified.

**It is well known that sampling rates for passive samplers are affected by temperature, pressure, humidity and wind speeds. In this study these parameters are ignored even- though the metrological conditions at the sampling location have been measured and are available.**

The reviewer is correct that in the current evaluation of the three passive air samplers, we have chosen to use sampling rates that are assumed invariant with respect to meteorological conditions, such as temperature, pressure, relative humidity, and wind speed. As stated in response to comments by reviewers #1 and #3, we decided to not consider the influence of meteorological parameters since the dependence of the sampling rate on temperature and wind speed has so far only been quantified extensively for one of the samplers, i.e. the *Mer*PAS® (McLagan et al., 2018a). However, as we reported on line 539, we observed a slight change of the SR.

**The automated Tekran analyzers are based on a standardized volume measurement and the passive samplers are not. If the authors ignore the metrological conditions then the comparison is not valid.**

This is not entirely correct. A Tekran analyzer does not measure the volume of air being sampled, but measures the mass of the air being sampled. In order to calculate volumetric air concentrations, that mass of air is converted into a volume of air applying the density of air at 0 °C and 1 atm. In a passive sampler, the sampled air volume is obtained from a sampling rate, which itself is the result of a calibration. What conditions apply to that volume, therefore, depend on the active sampling method that was used during the calibration. As in all cases a Tekran

analyzer was used during calibration of the PAS for Hg, the same conditions apply and the comparison is valid.

**At both sites the metrological conditions were highly variable and in addition to this no pressure correction has been made for barometric pressure and elevation on each site. The impact of metrological conditions on sampling rates is discussed later on in the paper as an explanation for the higher variability at the Canadian site so why was the volumetric correction not applied? I would prefer that all passive sampler results are reported using a standardized volume based on the average metrological conditions on site for the deployment period.**

As already indicated in the response to reviewer #3, one reason for assuming a constant sampling rate in our study is that the dependence of the sampling rate on temperature, wind speed, and relative humidity is not known for the IVL-PAS and CNR-PAS. It has only been quantified for the PAS developed by the University of Toronto on which the *Mer*PAS® is based (McLagan et al., 2018a). We have chosen to use invariant sampling rates in our comparison, and accordingly the assessment of bias and accuracy (e.g. Table 2, Figure 3) applies to a situation where a concentration obtained with a PAS while applying a constant sampling rate is compared with a concentration obtained with a Tekran.

**It would also be good to know how the uncertainty of the sampling rate is affected by the metrological conditions. Whilst the overall correction might be small it is still important when performing and report the comparison. Humidity is unlikely to have an impact. Wind speed could have an impact but since the samplers are housed in a shelter then one could argue that this could also be ignored. The correction should therefore focus on temperature and pressure.**

For the *Mer*PAS®, the dependence of the sampling rate on meteorological conditions is known quantitatively (McLagan et al., 2017, Snow et al., 2021a).

-   Laboratory experiments at different temperatures established that the impact of temperature on the sampling rate is small (it changes by 0.0009 $m^3$/day for 1 K change in temperature) and can be fully explained by the effect of temperature on the diffusivity of elemental mercury.

-   Laboratory experiments at different relative humidity found no effect on the sampling rate.

-   Laboratory experiments at different wind speed established that the sampling rate at wind speeds > 1 m/s the dependence of the sampling rate is minor (it changes by 0.003 $m^3$/day for 1 m/s change in wind speed). At wind speeds below 1 m/s, the dependence becomes more pronounced. A sampler evaluation involving numerous field sites with a wide range of meteorological conditions also lends support to a slight wind speed dependence of the sampling rate (McLagan et al. 2018b).

Barometric pressure affects the PASs in two ways that cancel each other out. At lower pressure, the air is thinner and we might expect a sampling rate that is reduced proportionally to the lower pressure. On the other hand, mercury diffuses faster at lower pressure, which would increase the sampling rate, again to an extent that is proportional to the lower pressure. Lowering and increasing the sampling rate to the same extent means that the two effects cancel each other out and no adjustment should be necessary for a comparison with Tekran data, which are based on 1 atm of atmospheric pressure (i.e. sea level). We also note that Rende and Toronto are close to sea level in elevation and no notable impact of atmospheric pressure can be expected.

We could add a discussion of the dependence of sampling rates on meteorological conditions to a revised version, but are not convinced that this is necessary, as what is currently known about this topic has been described in detail in earlier publications.

The CNR-PAS has a more recent history compared to the other samplers, and as thoroughly discussed, the influence of meteorological conditions on the sampling rate has not yet been fully investigated. Recently a paper dealing with the effect of meteorological conditions has been published (Avossa et al., 2020). Besides, although the results of the presented study are promising (in line 539 we reported "A promising result of this study is that the SRs of the CNR-PAS at the two locations are more similar than for the other two PASs, which may hint at an SR that has a relatively small dependence on meteorological factors…"), further tests aimed at calibrating the sampler under a wide variety of meteorological circumstances are still in progress.

**For the Accuracy assessment was performed by comparing the result the Tekran analyzers operating on each site. The paper explains that two Tekran instruments were operated but it seems that results from only one analyzer was used for the comparison? Surely, the comparison should be done on the average of the two Tekran results? If there are periods of known downtime for one of the two analyzers or technical justifications for ignoring a period of time then that is understandable but to ignore one analyzer and assume that the second analyzer is accurate does not seem correct.**

As mentioned both in the manuscript and in the response to reviewer #1, the Tekran system labelled "0075" "experienced an 8 % shift in the mass-flow meter calibration during the study" and "since it was not possible to determine when the shift occurred" (line 268), it is not a matter resolved by identifying instrument downtime or separating data from before or after the shift occurred.

**It seems that the authors have adopted a strategy of running analyzers in parallel but only using the second analyzer as reassurance them that the other analyzer was performing well. It would be nice to see the trend for both analyzers especially when so called spike events are experienced.**

The "0075" unit did not meet QA/QC criteria and we were unable to correct the data as addressed in previous comments. Therefore, it would be misleading to present the data as a time series alongside the "5037" unit results which met all QA/QC criteria. In this case, the redundancy of having two systems running in parallel allowed us to have at least one Tekran system generating data suitable for comparison with the PAS, and allowed us to ascertain confidence in the active measurement.

**Other concerns relate to the online analyzer reporting GEM even though the description of the analyzer suggests that TGM was determined because sampling was performed directly onto gold traps?**

As indicated in a response to reviewer #1, we refer to the analyte in the manuscript as gaseous mercury, i.e. we avoid the nomenclature of TGM and GEM. We compare what is measured by a Tekran system with what is measured by the PASs. The PASs were calibrated with the same type of Tekran system, so the sampling rates are clearly appropriate for that analyte. In a revised version, we can make sure to not use the term "GEM", but consistently refer to gaseous mercury throughout.

**The authors need to focus on what they are actually measuring with each technique applied. As mentioned above the Tekran measurements are based on a standardized volume whereas the passive samplers are not. It would be acceptable in the paper to discuss this in more detail and refer to comparability rather than accuracy. Having a passive sampler offering good agreement to the automated measurement is a very reassuring but to define the accuracy on this basis is not acceptable.**

We believe we have addressed these points already. The fact of the matter is that the concentrations obtained by Tekran and PASs are comparable, if only because the PASs' sampling rate have been previously calibrated with the help of Tekran systems and therefore do apply to the same conditions. We also state explicitly in the paper (line 381) that "Tekran values were considered as a benchmark for pragmatic reasons, knowing full well that this measurement itself may provide biased results".

**Overall this is very interesting study and the publication will be interest to many readers conducting mercury research in the field of atmospheric measurements. If passive samplers are deployed at more sites in the future then it is important to standardize the sample rate volumes especially when performing comparisons with automatic measurement systems.**

We appreciate the reviewer's endorsement of the merits of our study.

**References**

Avossa, J., F. De Cesare, P. Papa, E. Zampetti, A. Bearzotti, M. Marelli, N. Pirrone, A. Macagnano. Characteristics and performances of a nanostructured material for passive samplers of gaseous Hg. *Sensors,* **2020**, *20*, 6021.

McLagan, D.S., M.E.E. Mazur, C.P.J. Mitchell, F. Wania. Passive air sampling of gaseous elemental mercury: A critical review. *Atmos. Chem. Phys.* **2016**, *16*, 3061–3076.

McLagan, D. S., C. P. J. Mitchell, H. Huang, B. Abdul Hussain, Y. D. Lei, F. Wania. The effects of meteorological parameters and diffusive barrier reuse on the sampling rate of a passive air sampler for gaseous mercury. *Atmos. Meas. Tech.* **2017**, 10, 3651-3660.

McLagan, D.S., B. Abdul Hussain, H. Huang, Y.D. Lei, F. Wania, C.P.J. Mitchell. Identifying and evaluating urban mercury emission sources through passive sampler-based mapping of atmospheric concentrations. *Environ. Res. Lett.* **2018a**, *13*, 074008.

McLagan, D.S., C.P.J. Mitchell, A. Steffen, H. Hung, C. Shin, G.W. Stupple, M.L. Olson, W.T. Luke, P. Kelley, D. Howard, G.C. Edwards, P.F. Nelson, H. Xiao, G.-R. Sheu, A. Dreyer, H. Huang, B. Abdul Hussain, Y. D. Lei, I. Tavshunsky, F. Wania. Global evaluation and calibration of a passive air sampler for gaseous mercury. *Atmos. Chem. Phys.* **2018b**, *18*, 5905-5919.

Si, M., D. S. McLagan, A. Mazot, N. Szponar, B. Bergquist, Y. D. Lei, C. P.J. Mitchell, F. Wania. Measurement of atmospheric mercury over volcanic and fumarolic regions on the North Island of New Zealand using passive air samplers. *ACS Space Earth Chem.* **2020**, *4*, 2435-2443.

Snow, M.A., M. Feigis, Y. D. Lei, C. P. J. Mitchell, F. Wania. Development, characterization, and testing of a personal passive sampler for measuring inhalation exposure to gaseous elemental mercury. *Environ. Int.* **2021a**, *146*, 106264.

Snow, M.A., G. Darko, O. Gyamfi, E. Ansah, K. Breivik, C. Hoang, Y. D. Lei, F. Wania. Characterization of inhalation exposure to gaseous elemental mercury during artisanal gold mining and e-waste recycling through combined stationary and personal passive sampling. *Environ. Sci.: Processes Impacts* **2021**b, DOI: 10.1039/d0em00494d.

Stupple, G.W., D.S. McLagan, A. Steffen. In situ reactive gaseous mercury uptake on radiello diffusive barrier, cation exchange membrane and Teflon filter membrane during atmospheric mercury depletion events. *In Proceedings of the 14th International Conference on Mercury as a Global Pollutant*, Krakow, Poland, **2019**, 8-13 September.

Szponar, N., D. S. McLagan, R. Kaplan, C. P. J. Mitchell, F. Wania, A. Steffen, G. Stupple, B. Bergquist. Isotopic characterization of atmospheric gaseous elemental mercury by passive air sampling. *Environ. Sci. Technol.* **2020**, *54*, 10533-10543.